# Silver coordination-induced n-doping of PCBM for stable and efficient inverted perovskite solar cells

Cheng Gong [1,4], Haiyun Li[1,4], Huaxin Wang[1,4], Cong Zhang[1], Qixin Zhuang[1], Awen Wang[2], Zhiyuan Xu[1], Wensi Cai[1], Ru Li[1], Xiong Li [2] ✉ & Zhigang Zang [1,3] ✉

The bidirectional migration of halides and silver causes irreversible chemical corrosion to the electrodes and perovskite layer, affecting long-term operation stability of perovskite solar cells. Here we propose a silver coordination-induced n-doping of [6,6]-phenyl-$C_{61}$-butyric acid methyl ester strategy to safeguard Ag electrode against corrosion and impede the migration of iodine within the PSCs. Meanwhile, the coordination between DCBP and silver induces n-doping in the PCBM layer, accelerating electron extraction from the perovskite layer. The resultant PSCs demonstrate an efficiency of 26.03% (certified 25.51%) with a minimal non-radiative voltage loss of 126 mV. The PCE of resulting devices retain 95% of their initial value after 2500 h of continuous maximum power point tracking under one-sun irradiation, and > 90% of their initial value even after 1500 h of accelerated aging at 85 °C and 85% relative humidity.

Although the certified power conversion efficiency (PCE) of hybrid perovskite solar cells (PSCs) is ever increasing[1–4], their commercialization has been limited by the long-term operational instability[5–9]. It has been well-documented that the ion diffusion and chemical reactions between metal electrodes (i.e., Ag, Al or Cu) and perovskites contribute to the degradation of inverted (p-i-n) PSCs under built-in electric field or illumination[10–12]. Specially, irreversible changes such as metal migration and corrosion of electrode inflict far greater damage on the long-term operational stability of devices than the intrinsic degradation of the perovskite material itself[10]. Therefore, the operation stability of current inverted PSCs with high efficiency is less than 2000 h even operating at room temperature, and further drop to < 1500 h above 60 °C under maximum power point tracking (MPPT) and continuous illumination (Supplementary Table 1).

The Ag electrode-induced degradation in PSCs includes three primary mechanisms[13]: (1) the diffusion of volatile decomposition products or halide anions from the perovskite layer, such as halides or halogen species, into the Ag electrode, leading to metal corrosion and depletion of halides in the perovskite absorption layer; (2) the redox couple formation of metal contact with $Pb^{2+}$ ions in the perovskite film, accelerating the loss of halide and promoting the formation of $Pb^0$; (3) the diffusion of metals into the perovskite active layer under heat and/ or light activation, forming an insulating metal halide or defect state at the interface or bulk of the perovskite. Commonly, inert physical barriers, such as graphene[14], chromium or bismuth interlayers[8,15] and amorphous barrier films[11], are employed at the interface between the transport layer and the electrode to impede ion or metal diffusion. Despite, this strategy effectively delays metal electrode corrosion and prolongs device stability, iodine can still permeate through these barriers under heat or light conditions[16]. Alternatively, the chemical reaction between the metal electrode and perovskite that leads to electrode corrosion can be effectively mitigated by employing chemical anticorrosion methods involving materials capable of coordinating chemically with metals, using 1,3,5-triazine-2,4,6-trithiol

[1]Key Laboratory of Optoelectronic Technology & Systems (Ministry of Education), Chongqing University, Chongqing 400044, China. [2]Wuhan National Laboratory for Optoelectronics, Huazhong University of Science and Technology, Wuhan 430074 Hubei, China. [3]College of Information Science and Engineering, Yanshan University, Qinhuangdao 066004, China. [4]These authors contributed equally: Cheng Gong, Haiyun Li, Huaxin Wang. ✉e-mail: xiongli@hust.edu.cn; zangzg@cqu.edu.cn

trisodium salt (TTTS)[12], benzotriazole (BTA)[10] or calix[4]pyrrole (C[4] P)[17]. Although the integration of these barrier layers into PSCs can enhance their operational stability against environmental stressors, it offers limited improvement in terms of overall PCE, and may even come at the expense of PCE (Supplementary Table 2).

Herein, we propose a silver coordination-induced n-doping (CIN) of PCBM strategy using bipyridine derivative, i.e., 4,4′-dicyano-2,2′-bipyridine (DCBP). The DCBP with a heterocyclic structure exhibits pre-organized nitrogen coordination, enabling it to chelate Ag and release free electrons. These free electrons are subsequently absorbed by the typical electron acceptors PCBM, resulting in n-doping of PCBM. On the other hand, the DCBP can suppress the mutual migration of Ag and iodide as well as the formation of insulating AgI. As a result, the n-doped PCBM accelerates the electron extraction from perovskite layers, reducing accumulation of carriers at the interface, and remarkably suppressing the charge recombination. Finally, the resultant devices exhibit an efficiency of 26.03% (certified 25.51%) with a low non-radiative voltage loss of 126 mV, while the 1 cm$^2$ devices achieve a PCE of 24.17%. After continuous MPPT under one-sun illumination for 2500 h and under 85 °C and 85% relative humidity for 1500 h, the resulting devices retain approximately 95% and >90% of their initial efficiency, respectively.

## Results

### Protecting Ag electrodes and n-doping of PCBM via the CIN strategy

Under illumination and/or built-in electric field, perovskite-based optoelectronic devices are prone to ion or metal migration due to the intrinsic properties of perovskite materials[18]. In particular, irreversible degradations occur in PSCs when iodide ions migrate towards Ag electrodes and undergo reactions (Fig. 1a). Iodide ions migrating from the perovskite layer can corrode Ag electrodes over time, leading to the formation of insulating AgI and reducing conductivity[19,20]. Additionally, Ag can also migrate into the perovskite layer and accelerate the degradation of PSCs[21]. To overcome this issue, the DCBP molecules were introduced into the PCBM film. The PCBM films and related devices modified by DCBP are hereafter denoted as 'target'. First of all, we conducted density functional theory (DFT) calculations on DCBP molecules to obtain their electrostatic potential distribution, as well as molecular dipole and HOMO-LUMO distribution maps (Supplementary Fig. 1). The highly nucleophilic nitrogen atoms of pyridine are regarded as the primary binding sites for Ag. Simultaneously, there is a charge transfer between the iodine on the perovskite surface and the cyano group in DCBP, indicating a strong interaction between iodine and nitrogen[22,23]. DCBP molecules can effectively block the diffusion of Ag and prevent iodine intrusion into the electrode, as depicted in Fig. 1a. To this end, we conducted a more in-depth investigation with time-of-flight secondary ion mass spectrometry (TOF-SIMS) depth profiles, as shown in Fig. 1b, which illustrates the distribution of Ag and iodine in the devices following 800 h of one-sun exposure aging. The findings clearly reveal a substantial degradation in the control devices after long-term illumination, which we attributed to a significant augmentation of iodine content in both the electrode and the PCBM electron transport layer (ETL). Additionally, there is a notable elevation in Ag content within the perovskite absorption layer. In contrast, the variations in the internal content of iodine and Ag within the target devices after aging are exceedingly insignificant. This observation is consistent with the outcomes derived from the three-dimensional distribution maps (Supplementary Fig. 2) and mapping of TOF-SIMS signals of iodide and Ag on devices surface (Supplementary Fig. 3).

We further investigated the surface morphological changes of the Ag electrode before and after aging. The surface morphology of the Ag electrode in the control and target devices exhibit no significant differences in their initial states. The aged control devices exhibit large and deep holes on the Ag electrode, while the surface morphology of the Ag electrode in the aged target devices remains relatively uniform (Fig. 1c and Supplementary Fig. 4). X-ray diffraction (XRD) results confirm that in control devices significant AgI and PbI$_2$ are formed during aging, leading to substantial degradation of perovskite film crystallinity (Supplementary Fig. 5). In contrast, the aging process shows negligible AgI generation and limited PbI$_2$ formation in the target devices, with a relatively modest change in perovskite crystallinity. The diffusion of Ag electrodes and its impact on the perovskite layer are further demonstrated by cross-sectional scanning electron microscope (SEM) images (Supplementary Fig. 6). After aging, the Ag electrode is almost disappeared and exhibits a non-uniform cross-sectional morphology. Simultaneously, the perovskite layer also undergoes significant degradation. In contrast, the electrodes of the target devices are well-protected, with relatively minor changes in the perovskite layer, and noticeable suppression of the Ag and iodine migration. The above results conclusively demonstrate that DCBP molecules play a pivotal and positive role in inhibiting the Ag and iodine migration, which protects the Ag electrode and perovskite layer against damage and significantly prolongs the long-term stability of the PSCs.

Crucially, DCBP molecules, while coordinating with Ag, can simultaneously release electrons to n-doping PCBM, thus forming the CIN system, as confirmed by X ray photoelectron spectra (XPS) and auger electron spectra (AES)[24], which reveal the chemical state of Ag after coordination with DCBP. In Fig. 1d, Supplementary Fig. 7 and Supplementary Table 3, the binding energy and auger electron kinetic energy of standard Ag$^0$ reported in literature are documented as 368.2 eV and 357.9 eV[24], respectively. In this work, the binding energy and auger electron kinetic energy values for Ag 3$d$ of PCBM/Ag samples were found to be 368.2 eV and 357.9 eV, respectively, which are identical to those observed for standard Ag$^0$. This observation suggests that the valence state of Ag in the PCBM/Ag sample is zero (0). The valence state of Ag in compounds such as AgOOCCF$_3$ is +1, with corresponding binding energies and auger electron kinetic energies at 368.8 eV and 355.1 eV[24], respectively. The binding energy and auger electron kinetic energy values for Ag 3$d$ of PCBM@DCBP/Ag samples were found to be 368 .7 eV and 355.1 eV, respectively, which are identical to those observed for standard AgOOCCF$_3$. This observation suggests that the valence state of Ag in the PCBM@DCBP/Ag sample is zero (+1). Ag can coordinate with DCBP in PCBM, resulting in a change of its valence from 0 to +1. This alteration in valence signifies the loss of an electron by Ag, which is subsequently captured by PCBM to induce the n-doping effect. This can be further confirmed from time-of-flight mass spectrometry (Fig. 1e). It is evident that, in addition to the peak signals corresponding to the monomers of DCBP (206.8) molecules and PCBM (910.3), a prominent signal is observed at 313.5 ([Ag(DCBP)$^+$]) in the DCBP doped PCBM film, which indicates the formation of [Ag(DCBP)$^+$] within the PCBM film.

In the N 1$s$ spectra and Fourier transforms infrared (FTIR) spectra illustrated in Supplementary Fig. 8, the pyridine nitrogen and cyanide nitrogen undergo upward shifts of 0.71 and 0.16 eV in the high binding energy direction, respectively. These shifts indicate the capability of DCBP to chelate Ag, along with the strong interaction between cyanide group and iodine. Nuclear magnetic resonance hydrogen spectra ($^1$H NMR) further confirm the presence of coordination effect. The coordination interaction between DCBP and Ag significantly diminishes the electron density of the pyridine ring, resulting in an upward chemical shift of hydrogen atoms on the pyridine ring (Supplementary Fig. 9). The interaction between DCBP and iodine is further supported by the I 3$d$ XPS spectra and FTIR spectroscopy (Supplementary Fig. 10). To distinguish between the cyano and pyridine functional groups, molecules containing only cyano groups (1,3-Phenylenediacetonitrile, PhDT) and those containing only pyridine functional groups (1,10-Phenanthroline, Phen) are subjected to a series of characterizations

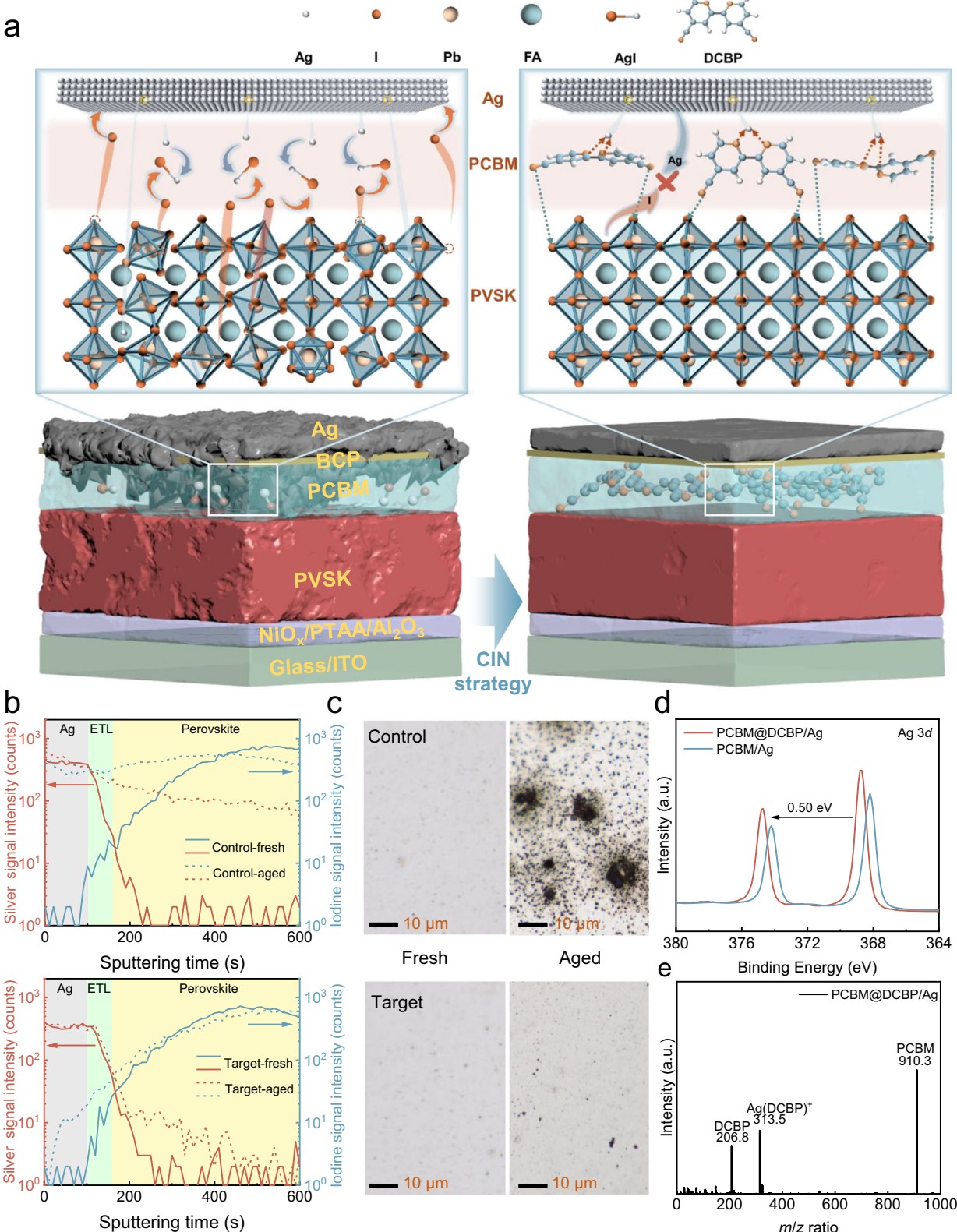

**Fig. 1 | The significant issue inherent in the devices and corresponding resolution. a** Schematic of the architecture of as-fabricated PSCs and the stabilizing strategy for the Ag electrodes and perovskite. **b** TOF-SIMS depth results of the iodine and Ag for control and target devices after continuous simulated AM1.5 illumination to aging for 800 h. **c** Optical microscope photographs of Ag electrodes, in their initial state and after continuous simulated AM1.5 illumination aging for 800 h. **d** The XPS spectra of Ag 3*d* are obtained for PCBM/Ag and PCBM@DCBP/Ag films after continuous simulated AM1.5 illumination to aging for 800 h. The thickness of the Ag electrodes is controlled below 5 nm in order to accurately measure the relationship between the electrode and the under layers. **e** Time-of-flight mass spectrum of PCBM@DCBP/Ag film after continuous simulated AM1.5 illumination to aging for 800 h. The Ag electrodes present were removed using Kapton tape after aging.

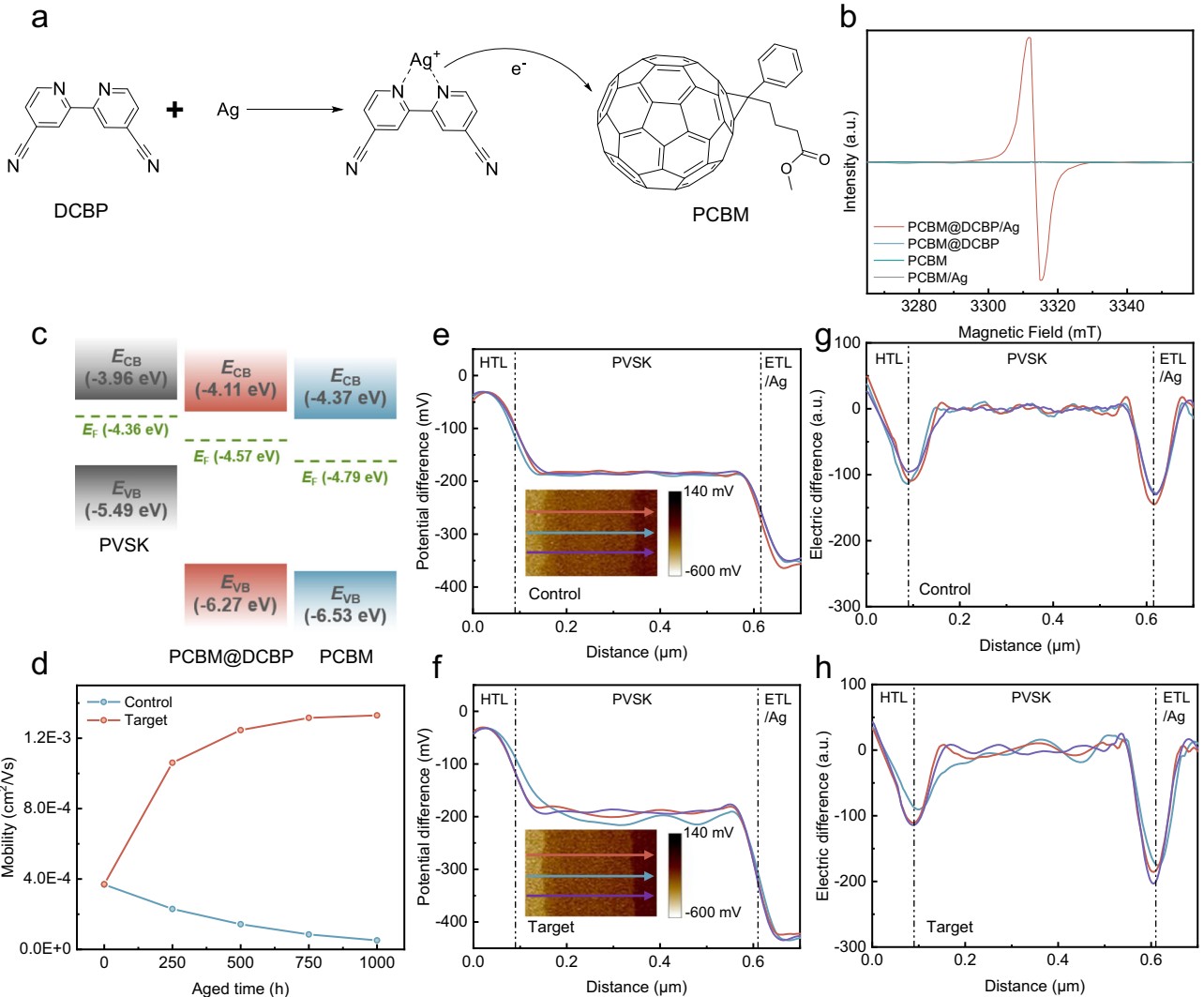

**Fig. 2 | Achieving n-type doping in the organic transport layer via coordination reaction. a** Reaction mechanism of the CIN strategy. **b** ESR spectra of PCBM, PCBM/ Ag PCBM@DCBP and PCBM@DCBP/Ag blend. **c** Energy level diagram of the PVSK, PCBM and PCBM@DCBP from Supplementary Fig. 23. **d** Changes of mobilities versus time for control (PCBM/Ag) and target (PCBM@DCBP/Ag) films under continuous simulated AM1.5 illumination. **e, f** Potential difference profiling across the devices under short-circuit conditions after continuous simulated AM1.5 illumination for 800 h. The insets show potential cross-sectional mapping. **g, h** Electric field in cross section of the devices.

(Supplementary Fig. 11). Through NMR, time-of-flight mass spectrum, UV-vis absorption, XPS, and FTIR characterization results, it is confirmed that the pyridine functional group can only coordinate with Ag, while the cyanide functional group can only strongly interact with I or FA (Supplementary Figs. 12–20). Furthermore, DFT simulations were conducted to understand the interaction between perovskite and DCBP. Here two representative defects, namely vacancies of FA ($V_{FA}$) and iodine ($V_I$) on the (001) FAI-terminated $FAPbI_3$ surface, interacting with DCBP unit are analyzed, as illustrated in Supplementary Fig. 21. The green clouds represent relative electron loss resulting from charge redistribution concerning bonds or atoms, while the yellow clouds indicate the electrons captured by respective atoms. Charge transfer occurs between iodine on the perovskite surface and the cyano group in DCBP, indicating a strong I-N interaction. This interaction restricts the movement/vibration of iodine, thereby suppressing potential formation of iodine vacancies. A strong interaction is also observed between the FA amine group in perovskite and the cyano group in DCBP, which limits the volatilization of FA and reduces the probability for formation of FA vacancies. These findings collectively confirm the successful implementation of the CIN strategy, achieving the goal of protecting the Ag electrode, suppressing ion migration, and thereby enhancing the stability of the devices.

## Effects of the CIN strategy

The reaction process of the CIN strategy is depicted in Fig. 2a, where the coordination of organic ligands and metals plays a crucial role in facilitating electron transfer, serving as a significant approach to achieve efficient Ohmic electron injection in organic photoelectrons[25,26]. Bipyridines and their derivatives possess heterocyclic structures with preorganized coordination sites for nitrogen, endowing them with strong chelating capabilities and exceptional electron transfer properties[27,28]. Therefore, they can readily form coordination complexes with metals, facilitating the release of free electrons due to an irreversible coordination reaction that shifts the equilibrium between metal atoms and cations in favor of the latter[28,29]. On the other hand, fullerenes and their derivatives are quintessential electron acceptor materials commonly utilized in organic optoelectronic devices[30–34]. Bipyridine derivatives can n-dope organic electron-transport layers through metal coordination. The introduction of DCBP into the PCBM film has no negative effect on the PCBM film morphology, as evidenced by Supplementary Fig. 22.

Electron spin resonance (ESR) measurements reveal a robust paramagnetic signal indicative of the generation of fullerene radical anions in the PCBM@DCBP/Ag blend (Fig. 2b). The n-doping enhances electron transfer in the fullerene ETL[35]. Furthermore, ultraviolet photoelectron spectroscopy (UPS) tests were conducted on PCBM and PCBM@DCBP films (Supplementary Fig. 23). The energy band alignment of PCBM films is also altered due to the increased electron concentration induced by n-doping effects. It is evident that the band alignment between the PCBM@DCBP/Ag film and the perovskite layer is more favorable (Fig. 2c and Supplementary Table 4), facilitating efficient electron transport while reducing accumulation of carriers at the interface, and hence significantly suppressing charge recombination.

The alteration in the electrical performance of PCBM films due to the CIN strategy is also notable (Fig. 2d and Supplementary Figs. 24–26). The mobility and electric conductivity of fresh PCBM/Ag and PCBM@DCBP/Ag films are almost on the same level, while they gradually decrease during the aging process in PCBM/Ag case. In contrast, the influence of the CIN strategy results in an increase in the electron mobility and electric conductivity of the aged PCBM@DCBP/ Ag films. During the illumination process, silver can coordinate with DCBP, leading to the formation of a coordination-induced n-doping (CIN) reaction in the [6,6]-phenyl C61 butyric acid methyl ester (PCBM) matrix. This n-doping effect introduces additional electrons into the PCBM, resulting in an increased electron concentration within the PCBM film. More electrons participate in the transport process, resulting in an increase in the conductivity of PCBM. In this work, the dopant concentration is low. Consequently, electron mobility is primarily affected by scattering due to lattice defects or impurity atoms at this stage. As electron concentration increases, the average distance between electrons decreases, leading to a reduction in collisions between electrons and lattice defects/impurity atoms[36–38]. Thus, this dynamic improves electron mobility by reducing the scattering effects of lattice defects or impurity atoms, facilitating more electrons movement within the semiconductor. With extended aging time under illumination, more Ag coordinates with DCBP, and PCBM obtains more electrons. This culminates in a progressive increment in electron concentration and a concomitant enhancement of electron mobility throughout the illumination period.

To further analyze the effect of the CIN strategy on the vertical electric field, the nanoscale kelvin probe force microscopy (KPFM) is utilized to map and observe the potential difference distribution across the devices cross-section under short-circuit conditions (Fig. 2e, f). Two notable descending trends at the HTL/perovskite (PVSK) and PVSK/ETL interfaces are observed, with no significant change in the depletion region width for both control and target devices[39,40]. Despite showing insignificantly effects on the potential drop between HTL and PVSK layer, the CIN strategy notably enhances the potential drop between ETL and PVSK layer, primarily attributed to n-type doping in the ETL. The first derivative of the potential difference is calculated to determine the distribution of the electric field across the cross-section of devices[41] (Fig. 2g, h). The electric field within the perovskite layer approaches zero, while the local electric field in the PVSK/ETL interface of the target devices are noticeably stronger than that of the control devices. A stronger local electric field facilitates charge collection and suppresses non-radiative charge recombination.

Although pyridine and its derivatives can form a CIN system with Ag, large molecular weight pyridine derivatives might tend to hinder charge transport in PSCs[24,25,29]. Therefore, we selected a series of pyridine derivatives and doped them into PCBM films at the same molar concentration. Both Bphen (4,7-Diphenyl-1,10-phenanthroline) and Phen (1,10-Phenanthroline) molecules reduced the conductivity of PCBM and performance of devices, highlighting the advantage of the molecular structure of DCBP as the smallest monomer among the pyridine derivatives (Supplementary Figs. 27–29).

Due to the strong interaction between DCBP and iodine or N–H (in FA), its presence on the surface of the PVSK can inhibit the formation of iodine/FA vacancies. Additionally, since DCBP chelate Ag and induce n-type doping in PCBM, the transport and extraction of charge carriers should be accelerated, as confirmed by time-resolved photoluminescence (TRPL) and photoluminescence (PL) measurements (Supplementary Figs. 30, 31 and Supplementary Table 5).

To further ascertain the passivation effect occurrence when DCBP is doped into PCBM, characterization via space charge limited current (SCLC) was performed. In Supplementary Fig. 32, the trap densities in perovskite films, using PCBM films both without and with DCBP, are quantified through SCLC measurements. The perovskite film modified by DCBP exhibits the lowest trap density of $1.60 \times 10^{15}$ cm$^{-3}$, while it is $2.26 \times 10^{15}$ cm$^{-3}$ in control film consistent with the results from PL and TRPL analyses. Relative to the control samples (1.22 µs), the target devices demonstrate a faster decay in photocurrent (0.48 µs) (Supplementary Fig. 33), indicating significantly improved carrier transport and extraction upon the incorporation of DCBP. As depicted in Supplementary Fig. 34, target devices exhibit an elevated built-in potential ($V_{bi}$) compared to the control devices, implying enhanced carrier separation and extraction owing to a more suitable band alignment. This observation is further cross-checked by light intensity-dependent results for open-circuit voltage ($V_{OC}$) (Supplementary Fig. 35). Control devices display an ideal factor of 1.51, whereas DCBP-modified PSCs exhibited a lower value of 1.27. This reduction can be attributed to the enhanced charge transport, along with the suppression of non-radiative recombination, contributing to improved $V_{OC}$ and fill factor (FF).

## Photovoltaic performance and stability

The optimal molar concentration of DCBP is $5 \times 10^{-3}$ mmol/mL, determined through the comparison of the photovoltaic performance of the devices. (Supplementary Figs. 36, 37). The relevant devices made from the addition of $5 \times 10^{-3}$ mmol/mL DCBP are hereafter denoted as 'target'. The photovoltaic performance of control and target PSCs are compared in both fresh and aged states to investigate the impact of the CIN strategy on the device performance during the aging process. In the fresh state, the target devices exhibit higher $V_{OC}$ and FF than the control, as depicted in Fig. 3a. The target devices architecture is composed of ITO/NiO$_x$/PTAA/AlO$_x$/perovskite/PCBM@DCBP/BCP/Ag. Initially, in the target devices, DCBP does not directly contact Ag (due to the presence of the cathode interlayer BCP), implying the absence of coordination-induced n-doping (CIN) reaction. Thus, as observed in Fig. 3a, compared to the control devices, the target devices exhibit only higher $V_{OC}$ and FF, attributed to the DCBP doped into PCBM effectively suppressing the formation of FA/I vacancies on the perovskite surface, thereby enhancing the quality of the perovskite film. However, during the illumination aging process, Ag can penetrate through the BCP layer into the PCBM layer, indicating a coordination reaction between DCBP and Ag, transferring free electrons to the strong electron acceptor PCBM, resulting in n-doping effects. The enhanced electrical performance of the PCBM layer facilitates electron extraction and transport, effectively reducing carrier accumulation at the interface, while non-radiative recombination at the interface is suppressed, leading to a significant reduction in charge losses. This is precisely the direct cause of the increased $V_{OC}$, $J_{SC}$ and FF of the devices in the aging process. Finally, the target devices achieve an efficiency of 26.03% after 800 h of continuous simulated AM1.5 illumination aging, with a stabilized PCE of 25.52% (Fig. 3b and Supplementary Fig. 38). The certified PCE was determined to be 25.51% with short-circuit current density ($J_{SC}$) = 26.24 mA/cm², $V_{OC}$ = 1.170 V, and FF = 83.1% (Supplementary Fig. 39). After aging, the performance of the control devices significantly deteriorates compared to their fresh states. Serious Ag and I interdiffusion occurs within the control devices, leading to severe non-radiative recombination and degradation of perovskite

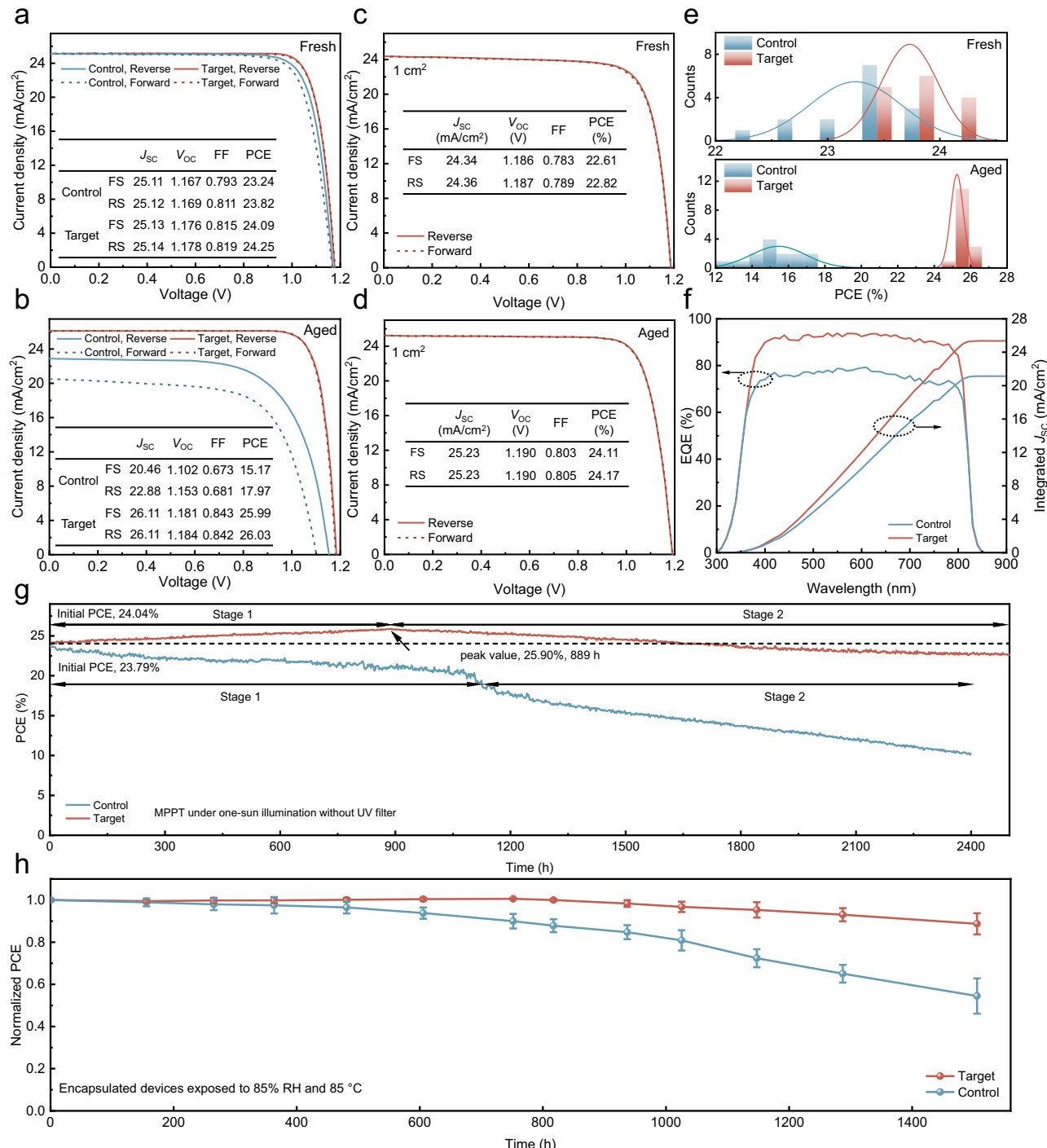

**Fig. 3 | Photovoltaic performance and stability.** *J−V* curves acquired during forward and reverse scans for both the champion control and target PSCs under **a** initial and **b** 800 h of continuous simulated AM1.5 illumination aging. Insets provide relevant photovoltaic parameters. *J−V* curves for the highest-performing control and target devices with a 1 cm² area under **c** initial and **d** 800 h of continuous simulated AM1.5 illumination aging. Insets present related photovoltaic parameters. **e** PCE histogram comparing the control and target PSCs under initial conditions and after 800 h of continuous simulated AM1.5 illumination aging. **f** EQE spectra of the control and target devices after 800 h of continuous simulated AM1.5 illumination aging. **g** PCE of the unencapsulated devices measured at MPPT. **h** The normalized PCE of encapsulated devices subjected to 85% RH and 85 °C in the dark with the error bar representing the standard deviation of four devices. Data are presented as mean values ± s.e.m.

layers/electrodes, greatly compromising the performance of the devices. (Fig. 3b)

A similar trend is observed in the target devices with a 1 cm² area. After 800 h of continuous simulated AM1.5 illumination aging, the efficiency of the target 1 cm² devices improve to 24.17% (Fig. 3c, d and Supplementary Fig. 40). the statistical distribution diagram demonstrates the excellent reproducibility of the target devices efficiency (Supplementary Figs. 41, 42 and Supplementary Tables 6, 7). After 800 h of continuous simulated AM1.5 illumination aging, the average PCE values of control devices experienced a decline from 23.25 ± 0.41% to 17.76 ± 1.50%. Conversely, the average PCE value of the target devices exhibited an enhancement from 23.73 ± 0.25% to

25.25 ± 0.34% (Fig. 3e). After aging, the target devices maintained a higher EQE intensity within the visible light range, while the control devices exhibited a significant decrease, as depicted in Fig. 3f. The integrated current densities of the control and target devices are 21.13 and 25.35 mA/cm², respectively. The aforementioned characterization methods serve as an evidence of the effectiveness of CIN strategies in enhancing and sustaining devices efficiency.

In Supplementary Fig. 43, the dark radiative recombination current ($J_{0, rad}$) calculation results in a radiative $V_{OC}$ limit of 1.310 eV. Consequently, there is an approximate non-radiative voltage loss of 126 mV concerning the $V_{OC}$ of the target devices (1.184 V). To find out the simultaneous enhancement of FF and $V_{OC}$ with the CIN system, we initially utilized photoluminescence quantum yield (PLQY) to evaluate non-radiative recombination in various components, such as the PVSK film, multi-layered partial cell stacks and complete devices. The non-invasive nature of PLQY characterization allows the separation of contributions from each layer/interface to non-radiative recombination losses. Different partial cell stacks are measured, and their results are compared with those of the complete devices. The $V_{OC}$ potential for each individual stack is quantified by the equation QFLS = $k_B T \times \ln(PLQY \times J_G/J_{0, rad})$, where $J_G$ is the generated current density at one-sun, calculated from PLQY results[42]. As displayed in Supplementary Fig. 44a and Supplementary Table 8, the QFLS of the PVSK increases from 1.205 eV to 1.226 eV after DCBP surface modification, indicating the passivation effect of DCBP molecules. For the PVSK/PCBM half-cell, the QFLS of the DCBP-treated PVSK/PCBM sample is 1.201 eV, remarkably close to the pristine perovskite, indicating that the interfacial recombination losses of PVSK/PCBM have been effectively overcome. Moreover, the close resemblance between QFLS and $V_{OC}$ in both control and target devices suggest uniformly flat Fermi levels and well-matched energy levels for majority carriers throughout the devices. Any misalignment, such as a downhill energy offset for electrons or an uphill offset for holes, when combined with a finite interface recombination velocity, may result in a gradient in quasi-Fermi levels for electrons or holes and a mismatch between QFLS and $V_{OC}$[43]. Supplementary Fig. 44b outlines the voltage loss mechanisms in control and target devices. Perovskite bulk and interface losses are the major contributors to devices voltage loss, with transport loss being minimal, as evidenced by the small differences in QFLS and $V_{OC}$ across the entire devices. The simultaneous suppression of perovskite bulk and interface losses by DCBP is a key factor in enhancing $V_{OC}$, thereby improving device efficiency.

Finally, we assessed the accelerated aging stability of the inverted PSCs. The stability of both unencapsulated control and target devices is determined through MPPT under consistent simulated AM1.5 illumination (100 mW/cm²) (45 °C). In Fig. 3g, the stability evolution of the target devices undergoes two stages: 1. The CIN system enhances the device efficiency while maintaining device stability, reaching a peak of 25.90% at 889 h; 2. The device performance gradually declines with a slower rate, maintaining 22.77% at 2500 h. For the control devices, under continuous illumination, the Ag electrode undergoes corrosion as iodine migrates out of the PVSK; Ag migrates into the perovskite layer, leading to irreversible damage to the PVSK. Ultimately, the PCE of the control devices exhibits only 11.51% after 2400 h. We further performed MPPT testing (ISOS-L-2) of the encapsulated device at elevated temperature (85 °C) (Supplementary Fig. 45)[44,45]. The target devices exhibit excellent high-temperature durability, maintaining >85% of its initial PCE even after continuous operation for 1000 h. In contrast, the PCE of control devices drop to only 6.7% after 1000 h of continuous high-temperature operation.

Figure 3h illustrates the damp-heat stability assessment conducted at 85 °C and 85% relative humidity for both the control and target devices. Over the course of 1500 h, the control devices experienced a degradation in PCE, reducing to 54.4% of their initial value. In contrast, the target devices exhibited a more favorable performance, with their PCE decreasing to 90% of the initial value. This outcome signifies enhanced damp-heat stability resulting from the implementation of the CIN strategy. The coordination reaction between DCBP and Ag gradually enhances the electrical properties of PCBM during the aging process, thereby accelerating the extraction of electrons from perovskite and effectively reducing non-radiative recombination. This phenomenon accounts for the performance improvement of the target device within the initial 889 h of aging. However, sustained external stimuli lead to continuous degradation within the device, ultimately resulting in a decline in device performance. The aging of the control devices is accelerated by the migration of Ag and iodine under external stimuli, ultimately resulting in diminished stability.

## Discussion
In summary, we chose the smallest monomer molecule, DCBP, from the pyridine family and incorporated it into the PCBM layer to suppress the notorious bidirectional migration of metal and iodine within the device. The chelation of DCBP with silver constructs a CIN system, releasing free electrons absorbed by the electron acceptor PCBM, inducing an n-doping effect. Additionally, DCBP exhibits strong interactions with iodine ions. The improvement of the silver electrode's anti-corrosion and the enhancement of the PCBM layer's electrical properties enable the fabrication of efficient and stable PSCs, particularly evident in their robust performance under prolonged exposure to light and heat stimuli. As a result, our CIN strategy leads to the resulting PSCs with a champion efficiency of 26.03% (certified 25.51%) and a low non-radiative voltage loss of 126 mV. After continuous MPPT under one-sun illumination for 2500 h and under 85 °C and 85% relative humidity for 1500 h, the resulting PSCs retain approximately 95% and >90% of their initial efficiency, respectively. This approach opens up avenues for enhancing the long-term stability and reliability of perovskite solar cells in practical applications.

## Methods
### Materials
The lead (II) iodide (PbI₂, 99.99%), cesium iodide (CsI, 99.99%), [6,6]-phenyl C₆₁ butyric acid methyl ester (PCBM), and formamidine hydroiodide (FAI, 99.5%) were obtained from Advanced Election Technology Co., Ltd. Nickel nitrate hexahydrate (Ni(NO₃)₂·6H₂O, 99.999%), sodium hydroxide (NaOH, 99.9%), N,N-dimethylformamide (DMF, 99.8%), isopropanol (IPA, 99.5%), dimethyl sulfoxide (DMSO, 99.9%), chlorobenzene (CB, 99.8%), and Al₂O₃ dispersed solution in IPA with a concentration of 20 wt.% were obtained from Sigma Aldrich. Poly(triaryl amine) (PTAA) with a molecular weight distribution of 6000-15000 and bathocuproine (BCP) were purchased from Xi'an Yuri Solar Corp., while the chemicals including: 4',4'-dicyano-2',2-bipyridine (DBCP, 97%), 4',7-diphenyl-1',10-phenanthroline (BPhen, 98%), and phenanthroline (Phen, 98%) were bought from Aladdin. Potassium hexafluorophosphate (KPF₆, 99%) was bought from Macklin. The NiOₓ nanoparticles (NPs) were synthesized based on previous research[46]. All chemicals and solvents used in this study were utilized without any additional purification.

### Device fabrication
The ITO-coated glass substrates were laser-etched, followed by ultrasonic cleaning of the etched ITO glass for 15 minutes using a detergent, deionized water, ethanol, and isopropanol in sequential order. The ITO-coated glass substrates were subjected to a 20 min treatment of ultraviolet-ozone (UVO). Subsequently, a NiOₓ NPs aqueous ink with a concentration of 25 mg/mL was prepared by dispersing the as-prepared NiOₓ NPs in deionized water. This ink was then spin-coated onto the ITO glass at a speed of 5000 rpm for 30 s. The NiOₓ films were annealed at 150 °C for 10 minutes, followed by immediate transfer into

a nitrogen-filled glove box. The $NiO_x$ films were spin-coated with a 2 mg/mL PTAA solution in CB at 6000 rpm for 30 s. Subsequently, the PTAA films were spin-coated with an $Al_2O_3$ dispersion solution (0.4 wt % in IPA) at 5000 rpm for 30 s. The $FA_{0.95}Cs_{0.05}PbI_3$ perovskite film was prepared by dissolving 228.4 mg of FAI, 18.2 mg of CsI, 645.4 mg of $PbI_2$, and 0.5 mg of $KPF_6$ in a mixed solvent solution ($v_{DMF}/v_{DMSO}$ = 4/1) with a concentration of 1.4 mmol/mL. The perovskite precursor solution was then spin-coated onto a glass/ITO/$NiO_x$/PTAA/$Al_2O_3$ substrate at speeds of 2000 rpm for 10 s followed by an additional spin at 4000 rpm for 40 s. During the second spin coating step, 150 µL of CB was deposited onto the perovskite film 5 seconds before the program ended. The resulting wet perovskite films were annealed at 100 °C for 30 minutes. Subsequently, a solution of PCBM in CB with a concentration of 23 mg/mL was spin-coated onto the perovskite films at a speed of 2500 rpm for 40 seconds. For the modified PCBM layer, DBCP, Bphen or phen with different molar concentration were added to the PCBM solution. Afterwards, 5 mg BCP was added into 1 mL IPA to prepare a supersaturated solution, which was filtered by PTFE filter before use. Afterward, the obtained saturated solution was spin-coated on PCBM film at 5000 rpm for 30 s. Finally, a thermal evaporation process under vacuum conditions ($2 \times 10^5$ Pa) was employed to deposit a Ag electrode with thickness around 100 nm.

## Characterization of the solar cells

The $J$–$V$ parameters of the PSCs were assessed in ambient air conditions (with a RH of 40–50%) employing a Newport-2612A solar simulator and a Keithley 2400 Source Meter. The illumination intensity of AM 1.5 G one sun (100 mW/cm$^2$) was rectified through an NIM calibrated standard Si solar cell. The active area of the cells was established as either 0.09 cm$^2$ and 1 cm$^2$. For EQE evaluation, a Newport Instruments system (Newport-74125) coupled with a lock-in amplifier and a 300 W xenon lamp was utilized.

## Characterization of the device stability

The stability of unencapsulated devices was assessed under continuous simulated solar illumination equivalent to one sun, emitted by a light-emitting diode (LED) lamp at 45 °C without a UV filter, at a controlled temperature of 25 ± 5 °C. MPPT was executed within a nitrogen-filled glovebox utilizing a dedicated MPPT system (YH-VMPP-S-16). Following the ISOS-T-1 standard, encapsulated devices were subjected to accelerated aging tests at 85 °C and 85% RH within a specialized climate chamber (ZK-301). Encapsulation was performed in a nitrogen atmosphere, involving the sealing of the device with a top glass cover and edge-sealing using a UV-curable adhesive. Curing of the adhesive was then accomplished through a two-minute exposure to ultraviolet light.

## PLQY

PLQY was measured using an integrating sphere (Fluorolog, Horiba JobinYvon), an Andor Kymera 193i spectrograph and a 660 nm continuous-wave laser (OBIS, Coherent) set at 1-sun equivalent photon flux (1.1 µm beam full-width half-maximum, 632 µW); photoluminescence was collected at normal incidence using a 0.1 numerical aperture, 110 µm-diameter optical fiber. For the calibration of the PLQY measurements we used a halogen lamp (HL-3 plus CAL from Ocean Optics).

## Other Measurements

The morphology of perovskite film was measured using field emission scanning electron microscopy (Apreo S HiVac FEI). XRD patterns were obtained with a Rigaku Ultima IV diffractometer equipped with Cu Kα radiation (λ = 1.5406 Å). Transient photocurrent measurements utilized a system excited by a 532 nm (1000 Hz, 6 ns) pulsed laser, and the photocurrent decay process was recorded with a 1 GHz Agilent digital oscilloscope (DSOX3102A) with a 50 X sampling resistor. Mott-Schottky measurements (1000 Hz) were conducted on a Chenhua electrochemical workstation (CHI 760E), and frequency-dependent capacitance measurements were performed on the same workstation. PL spectra and TRPL spectra were obtained by a fluorescence spectrophotometer (FLS1000). Raman mapping (LabRAM HR Evolution) was recorded under 532 nm excitation. TOF-SIMS measurements (PHI nanoTOF II Time-of-Flight SIMS) involved pulsed primary ions from an oxygen-ion beam (1 keV) for sputtering and a $Bi^+$ pulsed primary ion beam for analysis (25 keV). UPS, XPS, and AES measurements were conducted using a monochromatized Al source (Escalab Xi + ). XPS was calibrated using the peak position of C 1$s$, and UPS was calibrated using the work function of Au. $^1$H NMR data were collected from a Bruker 400 MHz NMR spectrometer. The preparation process of PCBM@DCBP/Ag samples in $^1$H NMR consists of two parts: first, a PCBM solution containing $5 \times 10^{-3}$ mmol/mL of DCBP is prepared into a thin film by spin-coating; then, thermally evaporating under vacuum ($2 \times 10^{-5}$ Pa) is employed to deposit Ag electrode layer with a thickness around 100 nm. The prepared samples are placed under continuous simulated AM1.5 illumination for 800 h. After that, the Ag electrodes on the surface of the samples are removed using Kapton tape. The sample film is separated from the substrate using a scraper and dissolved in the solution used for testing. Additionally, the preparation methods and aging conditions of the other two samples mentioned in Supplementary Fig. S9 remain consistent with those of the PCBM@DCBP/Ag samples. FTIR spectra were measured by Nicolet iS50 Infrared Fourier spectrometer. KPFM measurements were performed using a Bruker Dimension Icon (Germany) with AFM conducting tips featuring a resonance frequency ($\omega_0$) of approximately 140 kHz and a spring constant of 5.0 N/m. Standard AC mode imaging was employed for topography acquisition in the KPFM measurement. Short-circuit conditions were established by directly connecting the Ag electrode and the ITO electrode.

## DFT calculations

The geometry relaxation of our target molecule (4,4′-Dicyano-2,2′-bipyridine) on FAI terminated $FAPbI_3$ (001) surface ($3 \times 3 \times 1$ super-cell) was performed by open source CP2K package, the PBE-D3 functional with double-zeta basis sets (DZVP-MOLOPT) and Goedecker–Teter–Hutter pseudopotentials were used in the calculations, the energy cut off and real energy cut off were set to 400 Ry and 60 Ry, respectively. To avoid the image interaction, a vacuum layer of 15 Å was created in the z direction. We first put the target molecule above the $FAPbI_3$ surface at a distance of 3 Å, and then let the top three layer $FAPbI_3$ to relax until the maximum geometry change less than $3 \times 10^{-3}$ bohr, RMS geometry change less than $1.5 \times 10^{-3}$, maximum force smaller than $4.5 \times 10^{-4}$ hartree/bohr, and RMS force smaller than $3 \times 10^{-4}$. Charge density difference isosurface is exported to illustrate the electrostatic interaction. The electrostatic potential (ESP) mapping, molecule orbitals (HOMO, LUMO), molecule dipole, electronic affinity, and ionic potential were calculated by Gaussian 09 at the 6-311 + G (d, p) theoretical level. Open source tool Multiwfn was used to prepare the input file for cp2k calculations, and the ESP is also processed by the Multiwfn package.

## Reporting summary

Further information on research design is available in the Nature Portfolio Reporting Summary linked to this article.

## Data availability

Source data are provided with this paper. All the data supporting the findings of this study are available within this article and its Supplementary Information. Any additional information can be obtained from corresponding authors upon request. Source data are provided with this paper.

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

## Acknowledgements

This work was financially supported by the Defense Industrial Technology Development Program (JCKY2017110C0654) and the National Natural Science Foundation of China (62204028, 11974063, 62204026), Natural Science Foundation of Chongqing (CSTB2022NSCQ-MSX1514), China Postdoctoral Science Foundation (2022TQ0391, 2022M710507) and Chongqing Postdoctoral Science Special Foundation (2022CQBSHTB1026). We would like to thank the Analytical and Testing Center of Chongqing University for various measurements.

## Author contributions

C.G. conceived the idea and designed experiments. C.G., AW., Z.X., and C. Z. conducted the experiments and prepared films and devices. R.L. performed the DFT calculations. Z.Z., W. C, H.W., and C.G. designed and carried out the spectroscopy investigations and data analysis. Q.Z., C.G., and H.L. characterized thin film and devices. C.G. and H.L. fabricated the devices and performed certification. C.G., W. C., and Z. Z. prepared the first draft of the manuscript. C.G., X.L. and Z.Z. wrote the final version of the manuscript. All authors discussed the results and reviewed the manuscript Z.Z. supervised this project.

## Competing interests

The authors declare no competing interests.
