## [Peer Review File · Nature Communications]

Silver coordination-induced n-doping of PCBM for stable and efficient inverted perovskite solar cellsREVIEWER COMMENTS

Reviewer #1 (Remarks to the Author):

See Attachment

The study on inverted perovskite solar cells (PSCs) addresses a critical challenge in the field: the long-term operational stability compromised by chemical corrosion due to the bidirectional migration of halides and silver. The authors introduce an innovative approach using silver coordination-induced n-doping (CIN) of PCBM, aimed at protecting the Ag electrode and inhibiting iodine migration within the PSCs. By incorporating 4,4'-dicyano-2,2'-bipyridine (DCBP) into the PCBM layer, they effectively mitigate the mutual migration of detrimental species and enhance electron extraction, thereby improving device performance. The findings are in the interest of Nature Communications' readership, and we recommend publication after addressing the following queries.

(1) The descriptions in the manuscript on lines 60 and 163, pertaining to enhanced hole-blocking capability, are contentious and display inconsistencies with the information presented in Figure 2c and Table 4.

(2) While the authors extensively prove that devices exhibit improved performance after 800 hours of light aging compared to control devices, there is insufficient discussion on the mechanisms and factors contributing to the enhanced performance and reproducibility during this prolonged illumination period. For instance, What causes the enhancements in J_{sc} and FF? Furthermore, what leads to the increase in electron mobility?

(3) Are PVSK/DBCP@PCBM and PVSK/DBCP/PCBM the same description in Supplementary Fig. 30? If not, what is the difference?

(4) The authors mentioned that DCBP can not only n-dope PCBM to facilitate charge extraction but also chelate with Ag to inhibit the migration of metal and halide ions. Given these properties, can DCBP be directly used as a cathode interlayer, replacing BCP?

(5) In Supplementary Fig. 9, for preparing the ^1H NMR sample of PCBM@DCBP/Ag, typically, the sample is prepared using a solution method. However, the integration of Ag into the sample is a critical step that needs clear explanation. The author should describe how Ag is incorporated into the PCBM@DCBP matrix.

Reviewer #2 (Remarks to the Author):

This manuscript reported a silver coordination-induced n-doping (CIN) of PCBM strategy using bipyridine derivative (DCBP) to protect Ag electrode against corrosion and impede the migration of iodine within the PSCs. DCBP could effectively coordinate with both silver and iodide, leading to the suppression of mutual migration of Ag and iodide as well as the formation of insulating AgI. Meanwhile, the coordination between DCBP and Ag caused n-doping in PCBM layer, which was beneficial to the electron extraction from perovskite layers. Thus, the champion DCBP -doped inverted PSCs achieved an efficiency of 26.03% (certified 25.51%) with excellent operational and damp-heat stability. The manuscript was well prepared, and sufficient data was presented. Therefore, I would like to recommend publication of this manuscript in Nature Communication after the recommended improvements listed below have been implemented.

Comments of improvement:

1. How do the authors distinguish the different roles of two kinds of nitrogen? Can a coordination reaction occur between the cyano groups and Ag? The authors need to provide more substantial and persuasive evidence (expect XPS and FTIR measurements) to support this claim.
2. What does “Ag MNN kinetic energy” in Supplementary Table 3 mean? The authors need to provide a more detailed explanation of the Supplementary Table 3.
3. Authors should provide a reasonable explanation for the increase of the electron mobility (Fig 2d) and electric conductivity (Supplementary Fig. 17) on the aged PCBM@DCBP/Ag films.
4. Authors should provide the process of optimizing the doping concentration.
5. The device performance based on Bphen, Phen and DCBP should be compared.
6. In PLQY measurement, each sample needs to provide multiple PLQY values, along with the corresponding mean values and error bars.
7. Authors conducted the PLQY measurements on the PVSK/DCBP/PCBM sample, while DCBP was doped into the PCBM rather than an interface layer in the device. PLQY values on PVSK/PCBM@DCBP sample should be provided.
8. Authors believe that DCBP has passivation effect due to the QFLS calculation. However, the PVSK/PCBM@DCBP/Ag samples exhibited decreased PL intensity and PL lifetime compared to PVSK/PCBM/Ag samples in Supplementary Fig. 18. This suggested that DCBP could not exhibit passivation effect when doped into PCBM, which was contrary to the SCLC results in Supplementary Fig. 20. Please provide a reasonable explanation.
9. Was other passivator employed during the device fabrication? If DCBP could not exhibit passivation effect when doped into PCBM, it was incredible to obtain such a high VOC.
10. Ag would diffuse into the perovskite layer, which is particularly severe at high temperatures. Authors need to provide MPP data for the device at 85°C.

11. Page 6 line 113 “Fig. 1f” should be “Fig. 1e”.

Reviewer #3 (Remarks to the Author):

The manuscript reported a Ag-coordination doped PCBM for application in inverted PSCs with the introduction of DCBP additive into electrons transport layer. DCBP could coordinate with Ag and transport electrons to PCBM, inducing n-doping of PCBM. In addition, the author claimed that such coordination can also inhibit Ag migration toward perovskite and suppress its corrosion, thus enhancing device stability. Resulting device showed high efficiency of over 25% with good stability, retaining >90% of initial efficiency after MPP tracking for 2500 h. Despite much improved performance, there were still some issues needed to be resolved.

1. Inverted configuration of ITO/HTL/Perovskite/NCBP-PCBM/BCP/Ag was used here. Therefore, Ag electrode needed to migrate into PCBM layer to induce the doping of PCBM. Did such Ag migration induce extra degradation of PSCs? In addition, if Ag could easily migrate into PCBM to induce doping, how did this PCBM layer inhibit Ag migration? If such PCBM layer well inhibit Ag migration, how did the PCBM doping happen?

2. In previous works, Bi, rGO TTTS, or BTA were used to inhibit metal migration owing to their inert or compact film nature. While in this work, NCBP could react and coordinate with Ag electrode. It may confuse me why such reaction or coordination with Ag can inhibit Ag migration? From the viewpoint of chemistry, such reaction may promote the migration of Ag migration. There may be some other reasons for Ag migration inhibition, which needed to be further investigated.

3. In experimental section, the author claimed that BCP of 5mg/mL in IPA was used. However, such high concentration of BCP cannot dissolved in IPA.

4. The best performance appeared after long term aging (~800 h). During such aging period, perovskite degradation and PCBM improvement may co-exist. Will the device efficiency be better if we first doped PCBM with NCBP and Ag before spin-coating?

5. Besides Ag, can NCBP work in other metal electrodes, Cu, Al for example?

Response Letter

We acknowledge referees' insightful and professional comments and suggestions very much, which are valuable in improving the quality of our manuscript. Here we have addressed the queries from the reviewers point by point and revised the manuscript according to the comments. The revision was highlighted in red font in the revised manuscript.

Reviewer #1 (Remarks to the Author)

Comments: The study on inverted perovskite solar cells (PSCs) addresses a critical challenge in the field: the long-term operational stability compromised by chemical corrosion due to the bidirectional migration of halides and silver. The authors introduce an innovative approach using silver coordination-induced n-doping (CIN) of PCBM, aimed at protecting the Ag electrode and inhibiting iodine migration within the PSCs. By incorporating 4,4'-dicyano-2,2'-bipyridine (DCBP) into the PCBM layer, they effectively mitigate the mutual migration of detrimental species and enhance electron extraction, thereby improving devices performance. The findings are in the interest of Nature Communications' readership, and we recommend publication after addressing the following queries.

Reply: We thank the reviewer for positive evaluation and recognition on our work. We also appreciate the reviewer's valuable and constructive comments to help us improve the manuscript quality.

1. The descriptions in the manuscript on lines 60 and 163, pertaining to enhanced hole-blocking capability, are contentious and display inconsistencies with the information presented in Figure 2c and Table 4.

Reply: We gratefully appreciate the valuable comments. The fact, as emphasized by the reviewers, is that the assertion regarding the enhancement of hole-blocking capability is erroneous. We have made revisions to these inaccuracies in revised manuscript, as detailed below.

Revisions in the revised manuscript:

(1) Line 62, Page 4:

As a result, the n-doped PCBM accelerates the electron extraction from perovskite layers, **reducing accumulation of carriers at the interface**, and remarkably suppressing the charge recombination.

(2) Line 181, Page 10:

It is evident that the band alignment between the PCBM@DCBP/Ag film and the perovskite layer is more favorable (Fig. 2c and Supplementary Table 4), facilitating efficient electron transport while **reducing accumulation of carriers at the interface**, and hence significantly suppressing charge recombination.

2. While the authors extensively prove that devices exhibit improved performance after 800 hours of light aging compared to control devices, there is insufficient discussion on the mechanisms and factors contributing to the enhanced performance and reproducibility during this prolonged illumination period. For instance, what causes the enhancements in J_{sc} and FF? Furthermore, what leads to the increase in electron mobility?

Reply: We thank the reviewer for the professional comments. We have supplemented the revised manuscript with explanations and discussions regarding the mechanisms underlying the improvement of performance and repeatability during the aging process.

During the illumination process, silver can coordinate with DCBP, leading to the formation of a coordination-induced n-doping (CIN) reaction in the [6,6]-phenyl C61 butyric acid methyl ester (PCBM) matrix. This n-doping effect introduces additional electrons into the PCBM, resulting in an increased electron concentration within the PCBM film. More electrons participate in the transport process, resulting in an increase in the conductivity of PCBM. In this work, the dopant concentration is low. Consequently, electron mobility is primarily affected by scattering due to lattice defects or impurity atoms at this stage. As electron concentration increases, the average distance between electrons decreases, leading to a reduction in collisions between electrons and lattice defects/impurity atoms (*Nat. Rev. Mater.* 6, 531-549, 10.1038/s41578-021-00286-z; *Phys. Rev. B* 80, 195410, 10.1103/PhysRevB.80.195410; *Phys. Rev. B* 64, 195208, 10.1103/PhysRevB.64.195208). Thus, this

dynamic improves electron mobility by reducing the scattering effects of lattice defects or impurity atoms, facilitating more electrons movement within the semiconductor. With extended aging time under illumination, more Ag coordinates with DCBP, and PCBM obtains more electrons. This culminates in a progressive increment in electron concentration and a concomitant enhancement of electron mobility throughout the illumination period.

Throughout the aging process, the n-doped PCBM film and the perovskite layer achieve a more optimal energy band alignment owing to the ongoing CIN reaction. Concurrently, n-doping generates a more pronounced potential drop and intensifies the electric field at the perovskite/PCBM interface, which markedly expedites the extraction and transport of electrons. This enhancement effectively minimizes carrier accumulation, thus diminishing charge loss. Such improvements directly contribute to the augmented J_{SC} and FF observed in the devices during the aging process.

Meanwhile, control devices suffer from progressive degradation in both the perovskite and electrode materials over time, resulting in a broad range of performance inconsistency among the devices. Yet, the target devices exhibit pronounced resistance to the mutual migration of Ag and I throughout aging, mitigating performance fluctuation due to the migration-provoked charge transport irregularities and material deterioration. Consequently, the performance reproducibility of the target devices is enhanced. Furthermore, the superior energy band alignment and diminished interface recombination observed in the target devices during the aging process are also critical for the improved consistency in devices performance.

The manuscript has been supplemented with further discussions on the enhanced performance and underlying mechanisms of the target devices during its aging.

Revisions in the revised manuscript:

(1) Line 177, Page 9:

Furthermore, ultraviolet photoelectron spectroscopy (UPS) tests were conducted on PCBM and PCBM@DCBP films (Supplementary Fig. 23). **The energy band alignment of PCBM films is also altered due to the increased electron concentration induced by n-doping effects.** It is evident that the band alignment between the

PCBM@DCBP/Ag film and the perovskite layer is more favorable (Fig. 2c and Supplementary Table 4), facilitating efficient electron transport while **reducing accumulation of carriers at the interface**, and hence significantly suppressing charge recombination.

(2) Line 186, Page 10:

The mobility and electric conductivity of fresh PCBM/Ag and PCBM@DCBP/Ag films are almost on the same level, while they gradually decrease during the aging process in PCBM/Ag case. In contrast, the influence of the CIN strategy results in an increase in the electron mobility and electric conductivity of the aged PCBM@DCBP/Ag films. The mobility and electric conductivity of fresh PCBM/Ag and PCBM@DCBP/Ag films are almost on the same level, while they gradually decrease during the aging process in PCBM/Ag case. In contrast, the influence of the CIN strategy results in an increase in the electron mobility and electric conductivity of the aged PCBM@DCBP/Ag films. **During the illumination process, silver can coordinate with DCBP, leading to the formation of a coordination-induced n-doping (CIN) reaction in the [6,6]-phenyl C61 butyric acid methyl ester (PCBM) matrix. This n-doping effect introduces additional electrons into the PCBM, resulting in an increased electron concentration within the PCBM film. More electrons participate in the transport process, resulting in an increase in the conductivity of PCBM. In this work, the dopant concentration is low. Consequently, electron mobility is primarily affected by scattering due to lattice defects or impurity atoms at this stage. As electron concentration increases, the average distance between electrons decreases, leading to a reduction in collisions between electrons and lattice defects/impurity atoms³⁶⁻³⁸. Thus, this dynamic improves electron mobility by reducing the scattering effects of lattice defects or impurity atoms, facilitating more electrons movement within the semiconductor. With extended aging time under illumination, more Ag coordinates with DCBP, and PCBM obtains more electrons. This culminates in a progressive increment in electron concentration and a concomitant enhancement of electron mobility throughout the illumination period.**

(3) Line 249, Page 13:

The photovoltaic performance of control and target PSCs are compared in both fresh and aged states to investigate the impact of the CIN strategy on the devices performance during the aging process. In the fresh state, the target

devices exhibit higher V_{OC} and FF than the control, as depicted in Fig. 3a. The target devices architecture is composed of ITO/NiO_x/PTAA/AlO_x/perovskite/PCBM@DCBP/BCP/Ag. Initially, in the target devices, DCBP does not directly contact Ag (due to the presence of the cathode interlayer BCP), implying the absence of coordination-induced n-doping (CIN) reaction. Thus, as observed in Fig. 3a, compared to the control devices, the target devices exhibit only higher V_{OC} and FF, attributed to the DCBP doped into PCBM effectively suppressing the formation of FA/I vacancies on the perovskite surface, thereby enhancing the quality of the perovskite film. However, during the illumination aging process, Ag can penetrate through the BCP layer into the PCBM layer, indicating a coordination reaction between DCBP and Ag, transferring free electrons to the strong electron acceptor PCBM, resulting in n-doping effects. The enhanced electrical performance of the PCBM layer facilitates electron extraction and transport, effectively reducing carrier accumulation at the interface, leading to a significant reduction in charge losses. This is precisely the direct cause of the increased V_{OC} , J_{SC} and FF of the devices in the aging process. Finally, the target devices achieve an efficiency of 26.03% after 800 hours of continuous simulated AM1.5 illumination aging, with a stabilized PCE of 25.52% (Fig. 3b and Supplementary Fig. 38).

(4) Line 265, Page 13:

After aging, the performance of the control devices significantly deteriorates compared to their fresh states. Serious Ag and I interdiffusion occurs within the control devices, leading to severe non-radiative recombination and degradation of perovskite layers/electrodes, greatly compromising the performance of the devices. (Fig. 3b)

(5) Line 317, Page 16:

In contrast, the target devices exhibited a more favorable performance, with their PCE decreasing to 90% of the initial value. This outcome signifies enhanced damp-heat stability resulting from the implementation of the CIN strategy. The coordination reaction between DCBP and Ag gradually enhances the electrical properties of PCBM during the aging process, thereby accelerating the extraction of electrons from perovskite and effectively reducing non-radiative recombination. This phenomenon accounts for the performance improvement of the target devices within the initial 889 hours of aging. However, sustained external stimuli lead to continuous degradation within

the devices, ultimately resulting in a decline in devices performance. The aging of the control devices is accelerated by the migration of Ag and I under external stimuli, ultimately resulting in diminished stability.

References

36. Euvrard J, Yan Y, Mitzi DB. Electrical doping in halide perovskites. *Nat. Rev. Mater.* 6, 531-549 (2021).
37. Ha SD, Kahn A. Isolated molecular dopants in pentacene observed by scanning tunneling microscopy. *Phys. Rev. B* 80, 195410 (2009).
38. Maennig B, Pfeiffer M, Nollau A, Zhou X, Leo K, Simon P. Controlled p-type doping of polycrystalline and amorphous organic layers: Self-consistent description of conductivity and field-effect mobility by a microscopic percolation model. *Phys. Rev. B* 64, 195208 (2001).

3. Are PVSK/DCBP@PCBM and PVSK/DCBP/PCBM the same description in Supplementary Fig. 30? If not, what is the difference?

Reply: We thank the reviewer for the insightful comments. These two are different. In PVSK/DCBP@PCBM samples, DCBP exists as a dopant in PCBM, while in PVSK/DCBP/PCBM samples, it exists as an interfacial layer. In fact, DCBP is doped into the PCBM rather than an interface layer in the devices (Supplementary Fig. 44). Therefore, the PLQY values of PVSK/PCBM@DCBP are provided in the Supplementary Table 8, which closely resembles those of PVSK/DCBP/PCBM samples. We believe that the passivation of the perovskite layer by DCBP is similar, regardless of whether DCBP is present as an interlayer or as a dopant in PCBM.

Revisions in the revised supplementary information:

Supplementary Fig. 44. a, Photoluminescence quantum yield (PLQY) diagram for PVSK, PVSK/PCBM half stack, and full cell with/without DCBP. The nearly identical PLQY values observed in PVSK and PVSK/PCBM@DCBP samples suggest the mitigation of interfacial recombination. **The error bar representing the standard deviation of three samples. Data are presented as mean values \pm s.e.m.** **b**, Quasi-fermi level splitting (QFLS) and voltage loss mechanism for the control and DCBP-based samples.

Supplementary Table 8. PLQY and QFLS results of PVSK film, PVSK/PCBM half stack, and full cell with/without DCBP after 800 hours of continuous simulated AM 1.5G illumination aging. Data are presented as mean values \pm s.e.m for three samples.

Sample	PLQY (%)		QFLS (eV)	V_{oc} (V)
	Value	Mean values		
PVSK	1.64		1.205	
	1.76	1.66 ± 0.096		
	1.57			
PVSK/DCBP	3.82		1.227	
	3.90	3.84 ± 0.057		
	3.79			
PVSK/PCBM	0.505		1.175	
	0.635	0.530 ± 0.101		
	0.436			

	1.49			
PVSK/PCBM@DCBP	1.42	1.48 ± 0.056	1.202	
	1.53			
	1.45			
PVSK/DCBP/PCBM	1.53	1.49 ± 0.040	1.202	
	1.49			
	0.325			
cell-control	0.232	0.260 ± 0.059	1.157	1.151
	0.216			
	0.810			
cell-target	0.873	0.820 ± 0.054	1.187	1.184
	0.766			

4. The authors mentioned that DCBP can not only n-dope PCBM to facilitate charge extraction but also chelate with Ag to inhibit the migration of metal and halide ions. Given these properties, can DCBP be directly used as a cathode interlayer, replacing BCP?

Reply: We thank the reviewer for the constructive comments. Indeed, we have considered replacing BCP with DCBP as the cathode interlayer, aligning with the thoughts of the reviewer. However, DCBP can only dissolve in chlorobenzene solution, thus cannot be spin-coated onto the PCBM film as it would compromise the integrity of the PCBM layer. Given that DCBP is capable of chelating with Ag to form n-doping in PCBM while simultaneously suppressing metal/ion migration, it may serve a positive role as a cathode interlayer in facilitating the transport of charge carriers at the interface. However, if DCBP is used as a cathode interlayer, it avoids direct contact with the perovskite surface, thus limiting its potential to suppress FA/I vacancy defects on the perovskite surface. This, in turn, hampers the full exploitation of DCBP's capabilities. Therefore, we propose that doping DCBP into PCBM represents the optimal approach currently available.

5. In Supplementary Fig. 9, for preparing the ^1H NMR sample of PCBM@DCBP/Ag, typically, the sample is prepared using a solution method. However, the integration of Ag into the sample is a critical step that needs clear explanation. The author should describe how Ag is incorporated into the PCBM@DCBP matrix.

Reply: We thank the reviewer for the professional comments. The preparation process of PCBM@DCBP/Ag samples consists of two parts: first, a PCBM solution containing 5×10^{-3} mmol/mL of DCBP is prepared into a thin film by spin-coating; then, thermally evaporating under vacuum (2×10^{-5} Pa) is employed to deposit Ag electrode layer with a thickness around 100 nm. The prepared samples are placed under continuous simulated AM1.5 illumination for 800 hours. After that, the Ag electrodes on the surface of the samples are removed using Kapton tape. The film is separated from the substrate using a scraper and dissolved in the solution used for ^1H NMR testing.

Illumination ensures that Ag can migrate into the PCBM layer and is expected to undergo coordination reactions with DCBP, while the purpose of removing the silver electrode is to ensure that the sample used for ^1H NMR testing is completely soluble. Additionally, the preparation methods and aging conditions of the other two samples mentioned in Supplementary Fig. S9 remain consistent with those of the PCBM@DCBP/Ag samples. The preparation method for samples used in ^1H NMR is provided in the ‘Methods’ section of the revised manuscript, as follows.

Revisions in the revised manuscript:

Line 423, Page 21:

The preparation process of PCBM@DCBP/Ag samples in ^1H NMR consists of two parts: first, a PCBM solution containing 5×10^{-3} mmol/mL of DCBP is prepared into a thin film by spin-coating; then, thermally evaporating under vacuum (2×10^{-5} Pa) is employed to deposit Ag electrode layer with a thickness around 100 nm. The prepared samples are placed under continuous simulated AM1.5 illumination for 800 hours. After that, the Ag electrodes on the surface of the samples are removed using Kapton tape. The sample film is separated from the substrate using a scraper and dissolved in the solution used for testing. Additionally, the preparation methods and

aging conditions of the other two samples mentioned in Supplementary Fig. S9 remain consistent with those of the PCBM@DCBP/Ag samples.

Reviewer #2 (Remarks to the Author):

Comments: This manuscript reported a silver coordination-induced n-doping (CIN) of PCBM strategy using bipyridine derivative (DCBP) to protect Ag electrode against corrosion and impede the migration of iodine within the PSCs. DCBP could effectively coordinate with both silver and iodide, leading to the suppression of mutual migration of Ag and iodide as well as the formation of insulating AgI. Meanwhile, the coordination between DCBP and Ag caused n-doping in PCBM layer, which was beneficial to the electron extraction from perovskite layers. Thus, the champion DCBP -doped inverted PSCs achieved an efficiency of 26.03% (certified 25.51%) with excellent operational and damp-heat stability. The manuscript was well prepared, and sufficient data was presented. Therefore, I would like to recommend publication of this manuscript in Nature Communication after the recommended improvements listed below have been implemented.

Reply: We thank the reviewer for positive evaluation and recognition on our work. We also greatly appreciate the reviewer's valuable and constructive comments to help us improve the manuscript quality.

Comments of improvement:

1. How do the authors distinguish the different roles of two kinds of nitrogen? Can a coordination reaction occur between the cyano groups and Ag? The authors need to provide more substantial and persuasive evidence (expect XPS and FTIR measurements) to support this claim.

Reply: We thank the reviewer for the valuable comments. To distinguish between the cyano and pyridine functional groups, molecules containing only cyano groups (1,3-Phenylenediacetonitrile, PhDT) and those containing only pyridine functional groups (1,10-Phenanthroline, Phen) are subjected to a series of characterizations (NMR, time-of-flight mass spectrum, UV-vis absorption, XPS and FTIR) (Supplementary Fig. 11). Compared to the Phen sample, the chemical state of the hydrogen atoms on the pyridine ring of Phen in the PCBM+Phen+Ag sample has shifted (a: 0.01 ppm; b: 0.04 ppm; c: 0.06 ppm; d: 0.01 ppm) (Supplementary Fig. 12a). Additionally, compared to the Phen sample, the chemical state of the carbon atoms on the pyridine ring of Phen in the PCBM+Phen+Ag sample has also shifted (e: 0.02 ppm; f: 0.03 ppm) (Supplementary Fig. 12b). The

results of ^1H NMR spectra and ^{13}C NMR indicate that the pyridine functional group in Phen can effectively form coordination reaction with Ag, resulting in a reduction in the electron density around the hydrogen and carbon atoms on the pyridine ring, leading to the observed shift phenomenon. In comparing the ^1H NMR and ^{13}C NMR spectra of PhDT and PCBM+PhDT+Ag samples, it is observed that the chemical states of hydrogen and carbon atoms around the cyanide group remain unchanged (Supplementary Fig. 13). This indicates that there is no coordination reaction between the cyanide group in PhDT and Ag.

Supplementary Fig. 11. The molecular structures of PhDT and Phen.

Supplementary Fig. 12. a, ^1H NMR spectra of the pure Phen molecule and PCBM+Phen+Ag solutions. **b,** ^{13}C NMR spectra of the pure Phen molecule and PCBM+Phen+Ag solutions. Ag powder is added to the deuterated

reagent containing PCBM and Phen and stirred for six hours to allow complete reaction of Phen with Ag. The resulting solution is then filtered to obtain the PCBM+Phen+Ag solution.

Supplementary Fig. 13. a, ^1H NMR spectra of the pure PhDT molecule and PCBM+PhDT+Ag solutions. **b,** ^{13}C NMR spectra of the pure PhDT molecule and PCBM+PhDT+Ag solutions. Ag powder is added to the deuterated reagent containing PCBM and PhDT and stirred for six hours to allow complete reaction of PhDT with Ag. The resulting solution is then filtered to obtain the PCBM+PhDT+Ag solution.

The ^1H NMR and ^{13}C NMR spectra of Phen, Phen+FAI, and Phen+PbI₂ are shown in Supplementary Fig. 14. The results indicate that the chemical states of hydrogen and carbon atoms in Phen remain unchanged, suggesting no interaction between the iodine atoms in FAI/PbI₂ and Phen. However, when PhDT is mixed separately with FAI or PbI₂, there is a significant shift in the chemical states of the hydrogen and carbon atoms around the cyanide

group compared PhDT sample (Supplementary Fig. 15). This indicates that the cyano group in PhDT is capable of exerting strong interactions with FAI or PbI_2 .

Supplementary Fig. 14. a, 1H NMR spectra of the pure PhDT molecule, PhDT+FAI and PhDT+ PbI_2 solutions. **b,** ^{13}C NMR spectra of the pure PhDT molecule, PhDT+FAI and PhDT+ PbI_2 solutions.

Supplementary Fig. 15. a, ¹H NMR spectra of the pure PhDT molecule, PhDT+FAI and PhDT+PbI₂ solutions. **b,** ¹³C NMR spectra of the pure PhDT molecule, PhDT+FAI and PhDT+PbI₂ solutions.

Further, PCBM solutions containing Phen and PhDT are prepared into thin films via spin-coating method. Subsequently, approximately 100 nm of Ag layer is deposited onto the thin films through thermal evaporation. The films are then aged for 800 hours under AM1.5 illumination to ensure a sufficient amount of Ag participates in the coordination reaction. The Ag layer is peeled off by Kapton tape to facilitate the smooth progress of UV-visible spectroscopy testing on the PCBM thin film modified by Phen or PhDT. In Supplementary Fig. 16a, an additional peak appears at 430 nm in the PCBM@Phen/Ag film (Ag removed by Kapton tape after aging) compared to the PCBM@Phen film. Furthermore, the UV spectra of PCBM@Phen/Ag (Ag removed by Kapton tape after aging) are identical to those of PCBM@PhDT, with no additional peaks observed (Supplementary Fig. 16b). This suggests that the pyridine groups in Phen can coordinate with Ag, whereas the cyano group in PhDT do not exhibit coordination with Ag.

Supplementary Fig. 16. a, UV-vis absorption spectra of the PCBM@Phen film. **b**, UV-vis absorption spectra of the PCBM@PhDT film.

The time-of-flight mass spectrum of the PCBM@Phen/Ag and PCBM@PhDT/Ag samples is depicted in Supplementary Fig. 17. Three distinct peaks are observed in Supplementary Fig. 17a, corresponding to the Phen monomer, PCBM, and Ag(Phen)^+ , indicating the formation of new chelates resulting from coordination reactions between Ag and Phen within the thin film. Conversely, Supplementary Fig. 17b displays only two prominent peaks, attributed to PhDT monomer and PCBM, suggesting that the cyano group in PhDT do not engage in coordination reactions with Ag.

Supplementary Fig. 17. a, Time-of-flight mass spectrum of PCBM@Phen/Ag film after continuous simulated AM1.5 illumination to aging for 800 h. The Ag electrodes present were removed using Kapton tape after aging. **b**, Time-of-flight mass spectrum of PCBM@PhDT/Ag film after continuous simulated AM1.5 illumination to aging for 800 h. The Ag electrodes present were removed using Kapton tape after aging.

The N 1s XPS spectra of the PCBM@Phen/Ag sample can be divided into two peaks, corresponding to the N atoms in the pyridine functional group and the N atoms in the complex formed between the pyridine functional group and Ag (Supplementary Fig. 18a). Combined with Supplementary Fig. 18c, it can be observed that the XPS spectra of Ag 3d in the PCBM@Phen/Ag sample exhibit significant shifts compared with PCBM/Ag sample. This indicates that the pyridine functional group is capable of forming effective coordination reactions with Ag.

However, the N 1s XPS spectra of PCBM@PhDT/Ag sample show only one peak, and its binding energy position remains unchanged compared to the PCBM@PhDT sample (Supplementary Fig. 18b). Similarly, the Ag 3d XPS spectra of PCBM@PhDT/Ag sample also show no significant changes compared to the PCBM@PhDT sample (Supplementary Fig. 18d). This indicates that the cyano groups in PhDT do not interact with Ag.

Supplementary Fig. 18. **a**, XPS spectra of N 1s for the PCBM@Phen/Ag and PCBM@Phen films after continuous simulated AM1.5 illumination aging for 800 hours. **b**, XPS spectra of N 1s for the PCBM@PhDT/Ag and PCBM@PhDT films after continuous simulated AM1.5 illumination aging for 800 hours. **c**, XPS spectra of Ag 3d for the PCBM/Ag and PCBM@Phen/Ag films after continuous simulated AM1.5 illumination aging for 800 hours. **d**, XPS spectra of Ag 3d for the PCBM/Ag and PCBM@PhDT/Ag films after continuous simulated AM1.5 illumination aging for 800 hours. The Ag electrodes present in both of these post-aging configurations were removed using Kapton tape.

In PVS/K/Phen samples, the N 1s XPS spectrum exhibits two peaks, positioned at 398.65 eV and 398.28 eV respectively, corresponding to the N atoms in FA⁺ and the pyridine moiety of Phen (Supplementary Fig. 19a). The binding energy of N atoms in FA⁺ in PVS/K is located at 398.65 eV, while in Phen, the binding energy of N atoms in the pyridine functional group is at 398.28 eV. In addition, the I 3d XPS spectrum in PVS/K/Phen samples

shows no significant changes compared to PVSK samples (Supplementary Fig. 19c). This suggests that Phen does not form strong interactions with the components in PVSK.

In PVSK/PhDT samples, the N 1s XPS spectrum exhibits two peaks at 399.79 eV and 399.56 eV, corresponding to the N atoms in FA⁺ and PhDT, respectively (Supplementary Fig. 19b). The binding energy position of N atoms in FA⁺ in PVSK is 398.65 eV, whereas for the N atoms in the pyridine moiety of PhDT, it is 399.37 eV. On the other hand, significant changes are observed in the I 3d XPS spectrum of PVSK/PhDT samples compared to PVSK alone (Supplementary Fig. 19d). This indicates that the cyanide group in PhDT can form strong interactions with iodine and FA⁺.

Supplementary Fig. 19. **a**, XPS spectra of N 1s for the PVSK, PVSK/Phen and Phen films after continuous simulated AM1.5 illumination aging for 800 hours. **b**, XPS spectra of N 1s for the PVSK, PVSK/PhDT and PhDT films after continuous simulated AM1.5 illumination aging for 800 hours. **c**, XPS spectra of I 3d for the PVSK and PVSK/Phen films after continuous simulated AM1.5 illumination aging for 800 hours. **d**, XPS spectra of I

3d for the PVSK and PVSK/PhDT films after continuous simulated AM1.5 illumination aging for 800 hours. The Ag electrodes present in both of these post-aging configurations were removed using Kapton tape.

The FTIR characterization further validates the aforementioned conclusions (Supplementary Fig. 20). A noticeable shift of the pyridine functional group is observed in Phen/Ag samples compared to Phen samples; however, no significant variation is observed in the pyridine functional group in PVSK/Phen compared to Phen samples. Similarly, the cyano group shows no significant change in PhDT/Ag samples compared to PhDT samples, whereas a noticeable variation in the cyano group is observed in PVSK/PhDT compared to PhDT samples.

The FTIR characterization results further confirm that the pyridine functional group in Phen can coordinate with Ag without interacting with I. Conversely, the cyano group in PhDT cannot coordinate with Ag but interacts strongly with I or FA.

Supplementary Fig. 20. **a**, Fourier transforms infrared (FTIR) spectra of Phen and Phen/Ag films. **b**, FTIR spectra of PhDT and PhDT/Ag films. **c**, FTIR spectra of Phen and PVSK/Ag films. **d**, FTIR spectra of PhDT and

PhDT/Ag films. c, FTIR spectra of PhDT and PVSK/Ag films. The Ag electrodes present in above post-aging configurations were removed using Kapton tape.

2. What does “Ag MNN kinetic energy” in Supplementary Table 3 mean? The authors need to provide a more detailed explanation of the Supplementary Table 3.

Reply: Thanks for your valuable comments. Auger electron spectroscopy (AES) is an analytical technique in surface science and materials science. Electrons from the N shell transition to vacancies created when electrons from the M shell of an atom are excited to become free electrons by an electron beam, releasing energy in the process. This released energy then excites another electron from the N shell to become an electron, which is referred to as MNN auger electron. The term ‘Ag MNN kinetic energy’ denotes the kinetic energy possessed by auger electrons excited from the N shell of an Ag atom.

In Supplementary Table 3, the binding energy and auger electron kinetic energy of standard Ag⁰ reported in literature are documented as 368.2 eV and 357.9 eV, respectively (*Nat. Commun.* 10, 1-7, 10.1038/s41467-019-08821-x). In this work, the binding energy and auger electron kinetic energy values for Ag 3d of PCBM/Ag samples were found to be 368.2 eV and 357.9 eV, respectively, which are identical to those observed for standard Ag⁰. This observation suggests that the valence state of Ag in the PCBM/Ag sample is zero (0). The valence state of Ag in compounds such as AgOCCF₃ is +1, with corresponding binding energies and auger electron kinetic energies at 368.8 eV and 355.1 eV, respectively (*Nat. Commun.* 10, 1-7, 10.1038/s41467-019-08821-x). The binding energy and auger electron kinetic energy values for Ag 3d of PCBM@DCBP/Ag samples were found to be 368.7 eV and 355.1 eV, respectively, which are identical to those observed for standard AgOCCF₃. This observation suggests that the valence state of Ag in the PCBM@DCBP/Ag sample is zero (+1). Ag can coordinate with DCBP in PCBM, resulting in a change of its valence from 0 to +1. This alteration in valence signifies the loss of an electron by Ag, which is subsequently captured by PCBM to induce the n-doping effect.

Revisions in the revised manuscript:

(1) Line 110, Page 6:

Crucially, DCBP molecules, while coordinating with Ag, can simultaneously release electrons to n-doping PCBM, thus forming the CIN system, as confirmed by X ray photoelectron spectra (XPS) and auger electron spectra (AES)²⁴, which reveal the chemical state of Ag after coordination with DCBP. In Fig. 1d, Supplementary Fig. 7 and Supplementary Table 3, the binding energy and auger electron kinetic energy of standard Ag⁰ reported in literature are documented as 368.2 eV and 357.9 eV²⁴, respectively. In this work, the binding energy and auger electron kinetic energy values for Ag 3d of PCBM/Ag samples were found to be 368.2 eV and 357.9 eV, respectively, which are identical to those observed for standard Ag⁰. This observation suggests that the valence state of Ag in the PCBM/Ag sample is zero (0). The valence state of Ag in compounds such as AgOCCF₃ is +1, with corresponding binding energies and auger electron kinetic energies at 368.8 eV and 355.1 eV²⁴, respectively. The binding energy and auger electron kinetic energy values for Ag 3d of PCBM@DCBP/Ag samples were found to be 368.7 eV and 355.1 eV, respectively, which are identical to those observed for standard AgOCCF₃. This observation suggests that the valence state of Ag in the PCBM@DCBP/Ag sample is zero (+1). Ag can coordinate with DCBP in PCBM, resulting in a change of its valence from 0 to +1. This alteration in valence signifies the loss of an electron by Ag, which is subsequently captured by PCBM to induce the n-doping effect. This can be further confirmed from time-of-flight mass spectrometry (Fig. 1e).

Supplementary Table 3. The binding energy of Ag 3d and the kinetic energy of Ag MNN for different Ag samples²². PCBM/Ag and PCBM@DCBP/Ag film samples were subjected to 800 hours of continuous simulated AM1.5 illumination aging. The thickness of the Ag electrodes is controlled below 5 nm in order to accurately measure the relationship between the electrode and the under layers.

	Sample	Ag 3d binding energy (eV)	Ag MNN kinetic energy (eV)
Reference	Ag	368.2	357.9
	Ag ₂ SO ₄	368.3	354.7

	Ag ₂ O	368.4	350.6
	AgOOCFF ₃	368.8	355.1
Experiment	PCBM/Ag	368.2	357.9
	PCBM@DCBP/Ag	368.7	355.1

3. Authors should provide a reasonable explanation for the increase of the electron mobility (Fig 2d) and electric conductivity (Supplementary Fig. 17) on the aged PCBM@DCBP/Ag films.

Reply: We thank the reviewer for the professional comments.

During the illumination process, silver can coordinate with DCBP, leading to the formation of a coordination-induced n-doping (CIN) reaction in the [6,6]-phenyl C61 butyric acid methyl ester (PCBM) matrix. This n-doping effect introduces additional electrons into the PCBM, resulting in an increased electron concentration within the PCBM film. More electrons participate in the transport process, resulting in an increase in the conductivity of PCBM. In this work, the dopant concentration is low. Consequently, electron mobility is primarily affected by scattering due to lattice defects or impurity atoms at this stage. As electron concentration increases, the average distance between electrons decreases, leading to a reduction in collisions between electrons and lattice defects/impurity atoms (*Nat. Rev. Mater.* 6, 531-549, 10.1038/s41578-021-00286-z; *Phys. Rev. B* 80, 195410, 10.1103/PhysRevB.80.195410; *Phys. Rev. B* 64, 195208, 10.1103/PhysRevB.64.195208). Thus, this dynamic improves electron mobility by reducing the scattering effects of lattice defects or impurity atoms, facilitating more electrons movement within the semiconductor. With extended aging time under illumination, more Ag coordinates with DCBP, and PCBM obtains more electrons. This culminates in a progressive increment in electron concentration and a concomitant enhancement of electron mobility throughout the illumination period.

Revisions in the revised manuscript:

(1) Line 186, Page 10:

The mobility and electric conductivity of fresh PCBM/Ag and PCBM@DCBP/Ag films are almost on the same level, while they gradually decrease during the aging process in PCBM/Ag case. In contrast, the influence of the CIN strategy results in an increase in the electron mobility and electric conductivity of the aged

PCBM@DCBP/Ag films. During the illumination process, silver can coordinate with DCBP, leading to the formation of a coordination-induced n-doping (CIN) reaction in the [6,6]-phenyl C61 butyric acid methyl ester (PCBM) matrix. This n-doping effect introduces additional electrons into the PCBM, resulting in an increased electron concentration within the PCBM film. More electrons participate in the transport process, resulting in an increase in the conductivity of PCBM. In this work, the dopant concentration is low. Consequently, electron mobility is primarily affected by scattering due to lattice defects or impurity atoms at this stage. As electron concentration increases, the average distance between electrons decreases, leading to a reduction in collisions between electrons and lattice defects/impurity atoms³⁶⁻³⁸. Thus, this dynamic improves electron mobility by reducing the scattering effects of lattice defects or impurity atoms, facilitating more electrons movement within the semiconductor. With extended aging time under illumination, more Ag coordinates with DCBP, and PCBM obtains more electrons. This culminates in a progressive increment in electron concentration and a concomitant enhancement of electron mobility throughout the illumination period.

References

36. Euvrard J, Yan Y, Mitzi DB. Electrical doping in halide perovskites. *Nat. Rev. Mater.* 6, 531-549 (2021).
37. Ha SD, Kahn A. Isolated molecular dopants in pentacene observed by scanning tunneling microscopy. *Phys. Rev. B* 80, 195410 (2009).
38. Maennig B, Pfeiffer M, Nollau A, Zhou X, Leo K, Simon P. Controlled p-type doping of polycrystalline and amorphous organic layers: Self-consistent description of conductivity and field-effect mobility by a microscopic percolation model. *Phys. Rev. B* 64, 195208 (2001).

4. Authors should to provide the process of optimizing the doping concentration.

Reply: We thank the reviewer for the professional comment. The determination of optimal doping molar concentration is established by comparing the photovoltaic performance of devices modified with varying molar

concentrations of DCBP, as illustrated in Supplementary Figs. 36, 37. The optimal concentration of doping DCBP can be observed to be 5×10^{-3} mmol/ml.

Supplementary Fig. 36. *J*–*V* curves acquired during forward and reverse scans for both the champion control and target PSCs modified by different concentration DCBP.

Supplementary Fig. 37. Statistical distribution diagram of the photovoltaic parameters for the PSCs modified by different molar concentration (mmol/mL) of DCBP at their initial states. The statistical data were obtained from 15 individual cells for each kind of devices.

Revisions in the revised manuscript:

Line 244, Page 13:

The optimal molar concentration of DCBP is 5×10^{-3} mmol/mL, determined through the comparison of the photovoltaic performance of the devices. (Supplementary Fig. 36, 37). The relevant devices made from the addition of 5×10^{-3} mmol/ml DCBP are hereafter denoted as ‘target’. The photovoltaic performance of control and target PSCs are compared in both fresh and aged states to investigate the impact of the CIN strategy on the devices performance during the aging process.

5. The devices performance based on Bphen, Phen and DCBP should be compared.

Reply: We thank the reviewer for the professional comment. The devices performance based on BPhen, Phen, and DCBP has been supplemented in the Supplementary Figs. 28, 29, as shown below. Among these devices, those modified with DCBP exhibit the best performance.

Supplementary Fig. 28. a-d J - V curves acquired during forward and reverse scans for both the champion control (a), BPhen (b), Phen (c) and DCBP (d) modified PSCs.

Supplementary Fig. 29. Statistical distribution diagram of the photovoltaic parameters of the control and Bphen-, Phen- and DCBP-modified PSCs at their initial states. The statistical data were obtained from 15 individual cells for each kind of devices.

Revisions in the revised manuscript:

Line 213, Page 11:

Both Bphen (4,7-Diphenyl-1,10-phenanthroline) and Phen (1,10-Phenanthroline) molecules reduced the conductivity of PCBM and performance of devices, highlighting the advantage of the molecular structure of DCBP as the smallest monomer among the pyridine derivatives (Supplementary Figs. 27-29).

6. In PLQY measurement, each sample needs to provide multiple PLQY values, along with the corresponding mean values and error bars.

Reply: We thank the reviewer for the valuable suggestions. Three PLQY values are provided for each sample and the average is calculated (Supplementary Table 8). Error bars for PLQY are added to Supplementary Fig. 44.

Revisions in the revised supplementary information:

Supplementary Table 8. PLQY and QFLS results of PVSK film, PVSK/PCBM half stack, and full cell with/without DCBP after 800 hours of continuous simulated AM 1.5G illumination aging. Data are presented as mean values \pm s.e.m for three samples.

Sample	PLQY (%)		QFLS (eV)	V_{oc} (V)
	Value	Mean values		
PVSK	1.64	1.66 ± 0.096	1.205	
	1.76			
	1.57			
PVSK/DCBP	3.82	3.84 ± 0.057	1.227	
	3.90			
	3.79			
PVSK/PCBM	0.505	0.530 ± 0.101	1.175	
	0.635			
	0.436			
PVSK/PCBM@DCBP	1.49	1.48 ± 0.056	1.202	
	1.42			
	1.53			
PVSK/DCBP/PCBM	1.45	1.49 ± 0.040	1.202	
	1.53			
	1.49			
cell-control	0.325	0.260 ± 0.059	1.157	1.151
	0.232			
	0.216			
cell-target	0.810	0.820 ± 0.054	1.187	1.184
	0.873			
	0.766			

Supplementary Fig. 44. a, Photoluminescence quantum yield (PLQY) diagram for PVSK, PVSK/PCBM half stack, and full cell with/without DCBP. The nearly identical PLQY values observed in PVSK and PVSK/PCBM@DCBP samples suggest the mitigation of interfacial recombination. **The error bar representing the standard deviation of three samples. Data are presented as mean values \pm s.e.m.** **b**, Quasi-fermi level splitting (QFLS) and voltage loss mechanism for the control and DCBP-based samples.

7. Authors conducted the PLQY measurements on the PVSK/DCBP/PCBM sample, while DCBP was doped into the PCBM rather than an interface layer in the devices. PLQY values on PVSK/PCBM@DCBP sample should be provided.

Reply: We greatly appreciate the reviewer providing professional suggestions to enhance the quality of our work. The PLQY values on PVSK/PCBM@DCBP are provided in the Supplementary Table 8, which closely resembles those of PVSK/DCBP/PCBM samples. We believe that the passivation of the perovskite layer by DCBP is similar, regardless of whether DCBP is present as an interlayer or as a dopant in PCBM.

Revisions in the revised supplementary information:

Supplementary Table 8. PLQY and QFLS results of PVSK film, PVSK/PCBM half stack, and full cell with/without DCBP after 800 hours of continuous simulated AM 1.5G illumination aging. Data are presented as mean values \pm s.e.m for three samples.

Sample	PLQY (%)		QFLS (eV)	V_{oc} (V)
	Value	Mean values		
PVSK	1.64	1.66 ± 0.096	1.205	
	1.76			
	1.57			
PVSK/DCBP	3.82	3.84 ± 0.057	1.227	
	3.90			
	3.79			
PVSK/PCBM	0.505	0.530 ± 0.101	1.175	
	0.635			
	0.436			
PVSK/PCBM@DCBP	1.49	1.48 ± 0.056	1.202	
	1.42			
	1.53			
PVSK/DCBP/PCBM	1.45	1.49 ± 0.040	1.202	
	1.53			
	1.49			
cell-control	0.325	0.260 ± 0.059	1.157	1.151
	0.232			
	0.216			
cell-target	0.810	0.820 ± 0.054	1.187	1.184
	0.873			
	0.766			

8. Authors believe that DCBP has passivation effect due to the QFLS calculation. However, the PVSK/PCBM@DCBP/Ag samples exhibited decreased PL intensity and PL lifetime compared to PVSK/PCBM/Ag samples in Supplementary Fig. 18. This suggested that DCBP could not exhibit passivation effect when doped into PCBM, which was contrary to the SCLC results in Supplementary Fig. 20. Please provide a reasonable explanation.

Reply: Our own perspective aligns with the reviewer's professional comments. Any reduction in photoluminescence intensity when adding charge extraction layers to a perovskite absorber layer is detrimental

for the V_{oc} . Therefore, photoluminescence quenching under open circuit conditions cannot serve as a reliable indicator for effective charge transfer to the electron or hole transport layer.

We sincerely apologize for the oversight in not providing specific information on the preparation method of the PVS₂K/PCBM@DCBP/Ag and PVS₂K/PCBM/Ag samples in PL/TRPL measurement. In fact, both samples were prepared using ITO as the substrate and tested under short-circuit conditions, which were achieved by direct contact between the Ag electrode and ITO (Fig. R1). Under short-circuit conditions, any observed decrease in PL/TRPL intensity can be an effective indicator of carrier transport acceleration, since electrons and holes can recombine rapidly through the external circuit (*Adv. Energy Mater.* 10, 1904134, 10.1002/aenm.201904134). The reason for the decreased PL intensity and lifetime in the ITO/PVS₂K/PCBM@DCBP/Ag samples compared to the ITO/PVS₂K/PCBM/Ag samples is that the PCBM film modified by DCBP exhibits more efficient electron transport at a short circuit (Supplementary Fig. 30). Specifically, in the presence of a quenching layer, the reduction in PL intensity and lifetime due to charge carrier transport outweighs the increase caused by the DCBP passivation effect. Consequently, the PVS₂K/PCBM@DCBP/Ag samples ultimately demonstrate lower PL intensity and lifetime, which is also consistent with the previous work (*Joule* 5, 2148-2163, 10.1016/j.joule.2021.06.001).

To further ascertain the passivation effect occurring when DCBP is doped into PCBM, characterization via SCLC was performed. In fact, the structure of the sample in Supplementary Fig. 32 is ITO/SnO₂/PVS₂K/PCBM with or without DCBP/BCP/Ag, and it has not undergone any aging treatment. We apologize for the previous incorrect description and have made the correction in the revised manuscript. The presence of the BCP layer and the absence of aging treatment prevent DCBP from undergoing coordination reactions with Ag to form n-doping effects. It ensures that the trap densities of the sample are solely affected by the passivation effect of DCBP. This eliminates additional interfering factors in the calculation of film trap densities by excluding the impact of DCBP on the electrical properties of PCBM. Ultimately, DCBP also exhibits a defect passivation effect when doped into PCBM, as depicted in Supplementary Fig. 32.

Fig. R1. Schematic diagram of sample short circuit conditions.

Supplementary Fig. 30. a, PL and **b**, TRPL spectra of the corresponding perovskite films with or without PCBM as an electron quencher prepared on non-conductive glass after 800 hours of continuous simulated AM1.5 illumination aging. **Note that the TRPL and PL for samples with ETL were measured at a short circuit. Short-circuit conditions were established by directly connecting the Ag electrode and the ITO electrode.**

Supplementary Fig. 32. SCLC plots were obtained for the electron-only devices with the structure ITO/SnO₂/PVSK/PCBM/BCP/Ag, using PCBM films both without and with DCBP.

Revisions in the revised manuscript:

Line 221, Page 11:

To further ascertain the passivation effect occurring when DCBP is doped into PCBM, characterization via space charge limited current (SCLC) was performed. In Supplementary Fig. 32, the trap densities in perovskite films, using PCBM films both without and with DCBP, are quantified through SCLC measurements.

9. Was other passivator employed during the devices fabrication? If DCBP could not exhibit passivation effect when doped into PCBM, it was incredible to obtain such a high V_{oc} .

Reply: We thank the reviewer for the professional comment. As elucidated in Comment 8, DCBP also acts to passivate defects on the surface of perovskite layer when incorporated into PCBM films. There are strong I-N interactions between iodine on the perovskite surface and cyano groups in DCBP. This interaction restricts the movement/vibration of iodine, thereby suppressing potential formation of iodine vacancies. Moreover, a strong interaction is also observed between the FA amine group in perovskite and the cyano group in DCBP, which limits the volatilization of FA and reduces the probability for formation of FA vacancies. Additionally, PCBM films modified by DCBP exhibit a more favorable energy band alignment with perovskite films, accelerating

electron extraction while reducing electron accumulation at the interface. Ultimately, these factors result in the devices achieving such a high V_{oc} .

10. Ag would diffuse into the perovskite layer, which is particularly severe at high temperatures. Authors need to provide MPP data for the devices at 85 °C.

Reply: We thank the reviewer for the professional comment. We further performed MPPT testing (ISOS-L-2) of the encapsulated devices at elevated temperature (85 °C) in atmosphere (Supplementary Fig. 45) (*Science* 383, 6688, 10.1126/science.adj9602; *Nat. Energy* 5, 35-49, 10.1038/s41560-019-0529-5). The target devices exhibit excellent high-temperature durability, maintaining the PCE of 20.85% even after continuous operation for 1,000 hours. In contrast, the PCE of control devices drops to only 6.72% after 1,000 hours of continuous high-temperature operation. The MPPT data for the devices at 85°C has been added to the revised supplementary information.

Supplementary Fig. 45. MPPT of the control and target PSCs measured at 85 °C under simulated AM1.5G illumination (100 mW/cm²).

Revision in the revised manuscript:

Line 308, Page 15:

We further performed MPPT testing (ISOS-L-2) of the encapsulated devices at elevated temperature (85 °C)^{44, 45}. The target devices exhibit excellent high-temperature durability, maintaining >85% of its initial PCE even after

continuous operation for 1,000 hours. In contrast, the PCE of control devices drop to only 6.7% after 1,000 hours of continuous high-temperature operation.

Reference:

44. Khenkin MV, et al. Consensus statement for stability assessment and reporting for perovskite photovoltaics based on isos procedures. *Nat. Energy* 5, 35-49 (2020).

45. Tang H, et al. Reinforcing self-assembly of hole transport molecules for stable inverted perovskite solar cells. *Science* 383, 1236-1240 (2024).

11. Page 6 line 113 “Fig. 1f” should be “Fig. 1e”.

Reply: We thank the reviewer for the reminder. Fig. 1f has been replaced by Fig. 1e, with modifications as follows.

Revisions in the revised manuscript:

Line 123, Page 6:

This can be further confirmed from time-of-flight mass spectrometry (Fig. 1e). It is evident that, in addition to the peak signals corresponding to the monomers of DCBP (206.8) molecules and PCBM (910.3), a prominent signal is observed at 313.5 ($[\text{Ag}(\text{DCBP})^+]$) in the DCBP doped PCBM film, which indicates the formation of $[\text{Ag}(\text{DCBP})^+]$ within the PCBM film.

Reviewer #3 (Remarks to the Author)

Comments: The manuscript reported a Ag-coordination doped PCBM for application in inverted PSCs with the introduction of DCBP additive into electrons transport layer. DCBP could coordinate with Ag and transport electrons to PCBM, inducing n-doping of PCBM. In addition, the author claimed that such coordination can also inhibit Ag migration toward perovskite and suppress its corrosion, thus enhancing devices stability. Resulting devices showed high efficiency of over 25% with good stability, retaining >90% of initial efficiency after MPP tracking for 2500 h. Despite much improved performance, there were still some issues needed to be resolved.

Reply: We appreciate the reviewer's valuable comments and suggestions. We have carefully revised our manuscript according to the proposed suggestions. We sincerely hope that the revised contents and supplementary experiments satisfy you.

1. Inverted configuration of ITO/HTL/Perovskite/DCBP-PCBM/BCP/Ag was used here. Therefore, Ag electrode needed to migrate into PCBM layer to induce the doping of PCBM. Did such Ag migration induce extra degradation of PSCs? In addition, if Ag could easily migrate into PCBM to induce doping, how did this PCBM layer inhibit Ag migration? If such PCBM layer well inhibit Ag migration, how did the PCBM doping happen?

Reply: We gratefully appreciate the valuable comments. We designed a series of experiments to validate the actual migration of Ag within the devices. PSCs with structure of ITO/HTL/Perovskite/PCBM/BCP/Ag were fabricated and treated according to the procedure outlined in Fig. R2 to obtain samples A and B. The relevant devices made by adding 5×10^{-3} mmol/ml DCBP are hereinafter referred to as the 'target'. The relevant samples obtained by the no aging target devices through the processing method in Fig1 are hereinafter referred to as 'target sample A (dark)' and 'target sample B (dark)'; the relevant samples obtained by the aging target devices through the processing method in Fig1 are hereinafter referred to as 'target sample A (light aging)' and 'target sample B (light aging)'. The aging condition is continuous simulated AM1.5 illumination for 500 hours. Similarly, unmodified devices are hereinafter referred to as the 'control'. The samples obtained from no aging or aging

control devices by the method in Fig. R2 are marked as ‘control sample A (dark)’, ‘control sample B (dark)’, ‘control sample A (light aging)’ and ‘control sample B (light aging)’, respectively.

Fig. R2 Schematic representation of peeling the Ag electrode and washing the BCP and PCBM with isopropanol/chlorobenzene for XPS, XRD and SEM characterization.

In the XPS spectra of Ag $3d$ for both the control and target sample A, which have not undergone light/thermal aging, there are no discernible peaks (Fig. R3 a). However, after light exposure aging, distinct signal peaks are observed in the Ag $3d$ XPS spectra of both the control and target sample A. This indicates that, in the absence of external environmental stimuli (light/heat), the migration of Ag proceeds slowly enough that Ag species cannot be detected in PCBM. However, the migration of Ag to the PCBM layer is significantly accelerated under light/heat conditions, irrespective of whether it occurs in control or target devices.

In Fig R3 b, the Ag $3d$ XPS spectra of both the control and target sample B, which have not undergone light/thermal aging, show no discernible peaks. However, after aging, distinct signal peaks are observed in the Ag $3d$ XPS spectra of the control sample B, while no significant signal peaks are observed in the Ag $3d$ XPS spectra of the target sample B. We believe that DCBP molecules in the target devices form a complex with Ag, which significantly inhibits the migration of Ag.

Fig. R3 a, Schematic of the architecture of sample A and the XPS spectra of Ag 3d are obtained for control and target sample A. **b**, Schematic of the architecture of sample B and the XPS spectra of Ag 3d are obtained for control and target sample B.

To verify the influence of Ag migration on the perovskite layer, XRD and SEM characterizations were performed on sample B (Figs. R4 and R5). In the control sample B, after light aging, the intensity of the perovskite layer's diffraction peaks noticeably decreased, accompanied by the formation of PbI_2 . Conversely, in the target sample B, after light aging, there was no change in the intensity of the perovskite layer's diffraction peaks, and no formation of PbI_2 was observed. Additionally, SEM images revealed that in the control sample B, partial degradation occurred on the surface of the perovskite layer after light aging, whereas the surface morphology of the perovskite layer in the target sample B showed no significant changes after light aging. This indicates that under illumination conditions, the migration of Ag to the PCBM layer of the target devices does not cause additional damage to the perovskite layer.

Fig. R4 Schematic of the architecture of sample B and the XRD patterns of the control and target sample B.

Fig. R5 Schematic of the architecture of sample B and the SEM images of the control and target sample B.

Based on the results described above, we constructed a schematic diagram of Fig. R6. In the dark, the migration of Ag within control and target devices is exceedingly slow, to the extent that signals of Ag species are undetectable within PCBM. However, under light/thermal stimulation, Ag migration is notably accelerated towards both PCBM and PVSK layers. Within control devices, the migration of silver within the device leads to additional degradation of the perovskite layer. For target devices, Ag migrates solely into the PCBM layer, inducing a coordination-induced n-doping effect. As silver migrates into the PCBM layer, it forms complexes with DCBP molecules to impede further migration towards the perovskite layer. In target devices, Ag does not

adversely affect both the perovskite and PCBM layers. Therefore, although Ag can migrate into the PCBM layer, it does not introduce additional degradation to the target devices.

Fig. R6 The schematic representation the actual migration of Ag within the devices under dark and illuminated conditions.

2. In previous works, Bi, rGO, TTTS, or BTA were used to inhibit metal migration owing to their inert or compact film nature. While in this work, DCBP could react and coordinate with Ag electrode. It may confuse me why such reaction or coordination with Ag can inhibit Ag migration? From the viewpoint of chemistry, such reaction may promote the migration of Ag migration. There may be some other reasons for Ag migration inhibition, which needed to be further investigated.

Reply: We gratefully appreciate the valuable comments. In fact, the reason why TTTS or BTA have been used by previous researchers to inhibit metal migration is not due to its inertness or the nature of its compact film, but rather because of its reactive property, which enables it to coordinate with metals (Fig. R7) (*Energy Environ. Sci.* 15, 2154-2163, 10.1039/D1EE04022G; *Sci. Adv.* 6, eabd1580, 10.1126/sciadv.abd1580).

The distribution of $\text{Ag}(\text{DCBP})^+$ species in PCBM is supplemented in Figure R8, indicating a gradient distribution of $\text{Ag}(\text{DCBP})^+$ species in the cross-sectional direction of the PCBM layer in the aged target devices. The highest content of $\text{Ag}(\text{DCBP})^+$ species is observed at the interface in contact with Ag. Combining the reply in Comment 1, we suggest that the complex formed between Ag and DCBP can effectively inhibit further migration of Ag into the device interior.

Fig. R7 a, Devices structure and schematic of the BCP layer with TTTS (*Energy Environ. Sci.* 15, 2154-2163, 10.1039/D1EE04022G). **b**, Devices configuration and schematic diagram of BTA anticorrosion (*Sci. Adv.* 6, eabd1580, 10.1126/sciadv.abd1580).

Fig. R8 TOF-SIMS depth results of the iodine, Ag and $\text{Ag}(\text{DCBP})^+$ for target devices after continuous simulated AM1.5 illumination to aging for 800 hours.

3. In experimental section, the author claimed that BCP of 5 mg/mL in IPA was used. However, such high concentration of BCP cannot dissolved in IPA.

Reply: We agree with the reviewer's comments that BCP is not soluble at 5 mg/ml concentration in IPA. Here 5 mg BCP was added into 1 ml IPA to prepare a supersaturated solution, which was filtered by PTFE filter before use (Fig. R9). The relevant details of BCP deposition have been revised in Method.

Fig. R9 The supersaturated solution (5 mg/ml) of BCP (left) and the solution filtered using a PTFE filter (right).

Revisions in the revised manuscript:

Line 381, Page 20:

Afterwards, 5 mg BCP was added into 1 mL IPA to prepare a supersaturated solution, which was filtered by PTFE filter before use. Afterward, the obtained saturated solution was spin-coated on PCBM film at 5000 rpm for 30 s.

4. The best performance appeared after long term aging (~800 h). During such aging period, perovskite degradation and PCBM improvement may co-exist. Will the devices efficiency be better if we first doped PCBM with DCBP and Ag before spin-coating?

Reply: Thank you for your professional comments. A PCBM solution containing Ag and DCBP is stirred thoroughly for 6 hours to ensure full reaction between DCBP and Ag. Subsequently, the solution is filtered to ensure that no solid particles are present. The PCBM directly doped with DCBP and Ag is used to fabricate devices (hereinafter referred to as the 'target-2'), and the performance of the devices is evaluated (Fig. R10). The

PCE of target-2 is determined to be 26.00%, with a short-circuit current density (J_{sc}) of 26.07 mA/cm², V_{oc} of 1.182 V and FF of 84.4%. The devices prepared by doping with both DCBP and Ag before spin coating did not achieve a higher PCE compared to devices modified only with DCBP (hereinafter referred to as the ‘target-1’, 26.03%). We believe that during the initial 800 hours of aging in the target-1 device, the perovskite may not actually undergo significant degradation, insufficient to decrease the device's PCE. This could be attributed to the effective inhibition of Ag and I migration by DCBP in the target-1 device, thereby protecting the perovskite layer from damage. Consequently, the champion PCEs obtained from both methods are very close.

Although the devices prepared by these two methods exhibit similar PCEs, the target-2 devices demonstrate poorer operational stability compared to the target-1 devices. We further performed MPPT testing (ISOS-L-2) of the encapsulated devices at elevated temperature (85 °C) in atmosphere (Fig. R11) (*Science* 383, 6688, 10.1126/science.adj9602; *Nat. Energy* 5, 35-49, 10.1038/s41560-019-0529-5). The target-1 devices exhibit excellent high-temperature durability, maintaining the PCE of 20.85% even after continuous operation for 1,000 hours. However, after operating at high-temperature for 1,000 hours, the target-2 devices only retain a PCE of 12.39%. Due to pre-coordination with Ag within the target-2 devices, DCBP may cannot hinder the migration of Ag to the perovskite layer during operation, resulting in severe degradation of the perovskite layer.

Fig. R10 *J-V* curves acquired during forward and reverse scans for the champion PSCs, modified with PCBM doping achieved through the addition of DCBP and Ag directly before the spin-coating.

Fig. R11 MPPT of the control and target PSCs measured at 85 °C under simulated AM1.5G illumination (100 mW/cm²).

5. Besides Ag, can DCBP work in other metal electrodes, Cu, Al for example?

Reply: Thank you for your professional comments. The XPS characterization is used to demonstrate the interaction between DCBP and Cu/Al. In Fig. R12 and Table R1, the binding energy of standard Cu⁰ reported in literature are documented as 952.5 eV and 932.7 eV, respectively. The valence state of Cu in compounds such as CuO and CuCO₃ is +2, with corresponding binding energy at 953.3 eV and 933.5 eV; 954.5 eV and 934.7 eV, respectively. In this work, the binding energy of Cu 2p in PCBM/Cu samples is found to be identical to those observed for standard Cu⁰ (i.e., at 952.5 eV and 932.7 eV). This observation suggests that the valence state of Cu in the PCBM/Cu sample is zero (0). In addition, the binding energy of Cu 2p in PCBM@DCBP/Cu samples is at 952.8 eV and 933.0 eV. The variation in binding energy suggests that Cu may potentially coordinate with DCBP. However, the binding energy of Cu in PCBM@DCBP/Cu samples is not closely related to the binding energy of Cu 2p in CuO and CuCO₃. In PCBM@DCBP/Cu samples, Cu may still exist in elemental form, and a change in valence state may not occur.

Fig. R12 The XPS spectra of Cu 2p are obtained for PCBM/Cu and PCBM@DCBP/Cu films after continuous simulated AM1.5 illumination to aging for 800 hours. The thickness of the Cu electrodes is controlled below 5 nm in order to accurately measure the relationship between the electrode and the under layers.

Table R1 The binding energy of Cu 2p for different Cu samples. PCBM/Cu and PCBM@DCBP/Cu film samples were subjected to 800 hours of continuous simulated AM1.5 illumination aging. The thickness of the Cu electrodes is controlled below 5 nm in order to accurately measure the relationship between the electrode and the under layers.

Sample		Cu 2p binding energy (eV)	
Reference	Cu	952.5	932.7
	CuO	953.3	933.5
	CuCO ₃	954.5	934.7
Experiment	PCBM/Cu	952.5	932.7
	PCBM@DCBP/Cu	952.8	933.0

Similarly, in Fig. R13 and Table R2, the binding energy of standard Al⁰ reported in literature are documented as 72.6 eV. The valence state of Al in compounds such as AlCl₃ and Al₂O₃ is +3, with corresponding binding energy at 76.9 eV and 74.6 eV. In this work, the binding energy of Al 2p in PCBM/Al samples is found to be

identical to those observed for standard Al⁰ (i.e., at 72.6 eV). This observation suggests that the valence state of Al in the PCBM/Al sample is zero (0). On the other hand, the binding energy of Al 2*p* in PCBM@DCBP/Al samples is at 72.7 eV. The variation in binding energy suggests that Al may potentially coordinate with DCBP. However, the binding energy of Al in PCBM@DCBP/Al samples is not closely related to the binding energy of Al 2*p* in AlCl₃ and Al₂O₃. In PCBM@DCBP/Al samples, Al may still exist in elemental form, and a change in valence state may not occur.

Fig. R13 The XPS spectra of Al 2*p* are obtained for PCBM/Al and PCBM@DCBP/Al films after continuous simulated AM1.5 illumination to aging for 800 hours. The thickness of the Cu electrodes is controlled below 5 nm in order to accurately measure the relationship between the electrode and the under layers.

Table R2 The binding energy of Al 2*p* for different Al samples. PCBM/Al and PCBM@DCBP/Al film samples were subjected to 800 hours of continuous simulated AM1.5 illumination aging. The thickness of the Al electrodes is controlled below 5 nm in order to accurately measure the relationship between the electrode and the under layers.

	Sample	Al 2 p binding energy (eV)
Reference	Al	72.6
	AlCl ₃	74.9

	Al ₂ O ₃	74.6
Experiment	PCBM/Al	72.6
	PCBM@DCBP/Al	72.7

Based on the above results, it can be inferred that the valence states of Cu and Al in PCBM@DCBP/Cu or PCBM@DCBP/Al may be the same as those of Cu⁰/Al⁰. If the valence states of the metals remain unchanged, the phenomenon of electron transfer to PCBM may not occur, indicating that the n-doping effect may not occur. To further validate whether Cu or Al induces n-doping effects, we perform ESR characterization (Figure R14). When DCBP is combined with Cu and Al separately, no significant signal peaks are formed, demonstrating the absence of fullerene anion generation. Therefore, while Cu and Al can coordinate with DCBP, they not induce n-doping effect on PCBM.

Fig. R14 a, ESR spectra of PCBM, PCBM/Cu PCBM@DCBP and PCBM@DCBP/Cu blend. **b**, ESR spectra of PCBM, PCBM/Al PCBM@DCBP and PCBM@DCBP/Al blend.

REVIEWERS' COMMENTS

Reviewer #1 (Remarks to the Author):

The authors answered all my questions very well, and I suggest it could be published in Nature Communications.

Reviewer #2 (Remarks to the Author):

Upon reading the revised manuscript, I found the authors have well addressed all my questions and concerns. I have no further comments.

Reviewer #3 (Remarks to the Author):

The comments have been resolved in the new version and i have no more questions.It can be published in Nc.